# Synchronous activity patterns in the dentate gyrus during immobility

**Martin Pofahl[1], Negar Nikbakht[1], André N Haubrich[1], Theresa Nguyen[1], Nicola Masala[1], Fabian Distler[1], Oliver Braganza[1], Jakob H Macke[2], Laura A Ewell[1], Kurtulus Golcuk[1], Heinz Beck[1,3]***

[1]Institute for Experimental Epileptology and Cognition Research, University of Bonn, Bonn, Germany; [2]Machine Learning in Science, Cluster of Excellence "Machine Learning", University of Tübingen, Germany & Department Empirical Inference, Max Planck Institute for Intelligent Systems, Tübingen, Germany; [3]Deutsches Zentrum für Neurodegenerative Erkrankungen e.V, Bonn, Germany

**Abstract** The hippocampal dentate gyrus is an important relay conveying sensory information from the entorhinal cortex to the hippocampus proper. During exploration, the dentate gyrus has been proposed to act as a pattern separator. However, the dentate gyrus also shows structured activity during immobility and sleep. The properties of these activity patterns at cellular resolution, and their role in hippocampal-dependent memory processes have remained unclear. Using dual-color in vivo two-photon $Ca^{2+}$ imaging, we show that in immobile mice dentate granule cells generate sparse, synchronized activity patterns associated with entorhinal cortex activation. These population events are structured and modified by changes in the environment; and they incorporate place- and speed cells. Importantly, they are more similar than expected by chance to population patterns evoked during self-motion. Using optogenetic inhibition, we show that granule cell activity is not only required during exploration, but also during immobility in order to form dentate gyrus-dependent spatial memories.

*For correspondence:
heinz.beck@ukbonn.de

## Introduction

The dentate gyrus receives polymodal sensory information from the entorhinal cortex, and relays it into the hippocampal network. The most prevalent view of the dentate gyrus input-output transformation in this circuit is that it acts as a pattern separator. This capability requires the animal to generate dissimilar neuronal representations from overlapping input states that represent similar but not identical environments (*Cayco-Gajic and Silver, 2019*). Such an operation, termed pattern separation, has been ascribed to the hippocampal dentate gyrus in species ranging from rodents to humans (*Berron et al., 2016*; *Leutgeb et al., 2007*; *Sakon and Suzuki, 2019*). In the dentate gyrus, polysensory inputs are mapped onto a large number of granule cells which exhibit extremely sparse firing patterns, resulting in a high probability of non-overlapping output patterns (*Danielson et al., 2016*; *GoodSmith et al., 2017*; *Hainmueller and Bartos, 2018*; *Pilz et al., 2016*; *Senzai and Buzsáki, 2017*; *van Dijk and Fenton, 2018*). This concept has been influential in understanding dentate gyrus function when processing multimodal, current, or 'online' sensory information during mobility and exploration.

However, the dentate gyrus is far from silent during immobility. It displays prominent electrographic activity patterns such as dentate spikes and sharp waves, which occur primarily during immobility or sleep (*Bragin et al., 1995*; *Meier et al., 2020*; *Penttonen et al., 1997*). In downstream hippocampal regions such as CA1, neuronal activity during immobility incorporates the replay of behaviorally relevant sequences during sharp wave ripples, a process important in memory consolidation (*Davidson et al., 2009*; *Diba and Buzsáki, 2007*; *Dupret et al., 2010*; *Foster and Wilson,*

*2006*; *Girardeau et al., 2009*; *Malvache et al., 2016*; *Skaggs and McNaughton, 1996*; *Wilson and McNaughton, 1994*). In the dentate gyrus, little detail is known about how granule cells are active during immobility at the population level, and it is unknown whether activity during immobility reiterates behaviorally relevant information. Moreover, the role of dentate gyrus activity during immobility in memory formation is unclear.

Here, we have used dual-color two-photon in-vivo $Ca^{2+}$ imaging to show that in immobile mice, the dentate gyrus exhibits frequent, sparse, and synchronous population events that at the population level are similar to activity patterns during locomotion. Moreover, we have tested the idea that dentate gyrus activity during immobility is relevant for dentate-gyrus dependent spatial memory.

## Results

### Sparse, structured dentate network events in immobile animals

We imaged the activity of large populations of hippocampal dentate granule cells (GCs) using a Thy1-GCaMP6s mouse line (GP4.12Dkim/J, *Dana et al., 2014*). In addition, we monitored the bulk activity of the major input system into the dentate gyrus, the medial perforant path (MPP). To this end, we expressed the red-shifted $Ca^{2+}$ indicator jRGECO1a (*Dana et al., 2016*) in the medial entorhinal cortex using viral gene transfer (see Materials and methods section, *Figure 1A*, *Figure 1—figure supplement 1A*). To allow efficient excitation of both genetically encoded $Ca^{2+}$ indicators, we established excitation with two pulsed laser sources at 940 and 1070 nm (see *Figure 1—figure supplement 1B–F*). The mice were placed under a two-photon microscope and ran on different variants of a linear track, equipped with different types of cues (see *Figure 1—figure supplement 1G–J*, *Figure 1—video 1*).

As previously described, the firing of GCs was generally sparse (*Danielson et al., 2016*; *Hainmueller and Bartos, 2018*; *Neunuebel and Knierim, 2012*; *Pilz et al., 2016*), both when animals were immobile and when they were running on a textured belt without additional cues (mean event frequency 1.38 ± 0.19 events/min and 0.97 ± 0.2 events/min, respectively, n = 1415 granule cells in nine mice, *Figure 1B*, *Figure 1—figure supplement 2C–E*). Despite the sparse activity of granule cells, we observed synchronized activity patterns (*Figure 1—video 2*). To rigorously define such events, we used an algorithm that detects synchronized network events within a 200 ms time window, corresponding to 1 ± 1 frame at our sampling rate (see Materials and methods). Such synchronous network events could readily be observed in the dentate gyrus in all mice (*Figure 1C,D*, network events depicted in different colors, see examples for ΔF/F traces of participating cells in *Figure 1E*). Network events were sparse, incorporating on average only 5.7 ± 0.09% of the total active GC population. Shuffling analysis (see Materials and methods) confirmed that network events do not arise by chance (*Figure 1F*, gray bars correspond to shuffled data, n = 9 mice, three sessions/mouse). This was robust over three different types of shuffling analysis (*Figure 1G*, *Figure 1—figure supplement 3* and Materials and methods for the description of the shuffling methods).

Notably, network events occurred mainly during immobility periods and were much less prevalent during running (*Figure 1D*, quantification in *Figure 1G*). Accordingly, network event frequencies were significantly higher during immobility (repeated measures ANOVA, $F_{(1,8)}=117$, $p=2\times10^{-6}$, n = 9 mice, three sessions, *Figure 1H*). During immobility periods (defined as running speeds < 4 cm/s), the vast majority of (99.6%) network events occurred when mice were completely immobile. Network events were on average evenly distributed during the 20 min imaging session (*Figure 1I*), as well as during individual periods of immobility (*Figure 1J*).

### Dentate network events are correlated with MPP activation

We then analyzed the activity in the MPP input fiber tract expressing jRGECO, and probed the relation of this activity with GC activity patterns. As expected during exploratory states, the bulk MPP activity was increased during locomotion, consistent with increased sensory input (*Figure 2A,B*, red channel, *Figure 2C* for average value). During immobility in particular, larger fluctuations of bulk MPP activity were observed. This phenomenon was reflected in a larger variance of the bulk MPP signal during immobility (*Figure 2D*). Cross-correlation revealed that during immobility, the increases in bulk MPP activity were associated with peaks in average GC activity levels (*Figure 2E*). Both signals were significantly correlated in most sessions for periods of immobility (8/9 sessions,

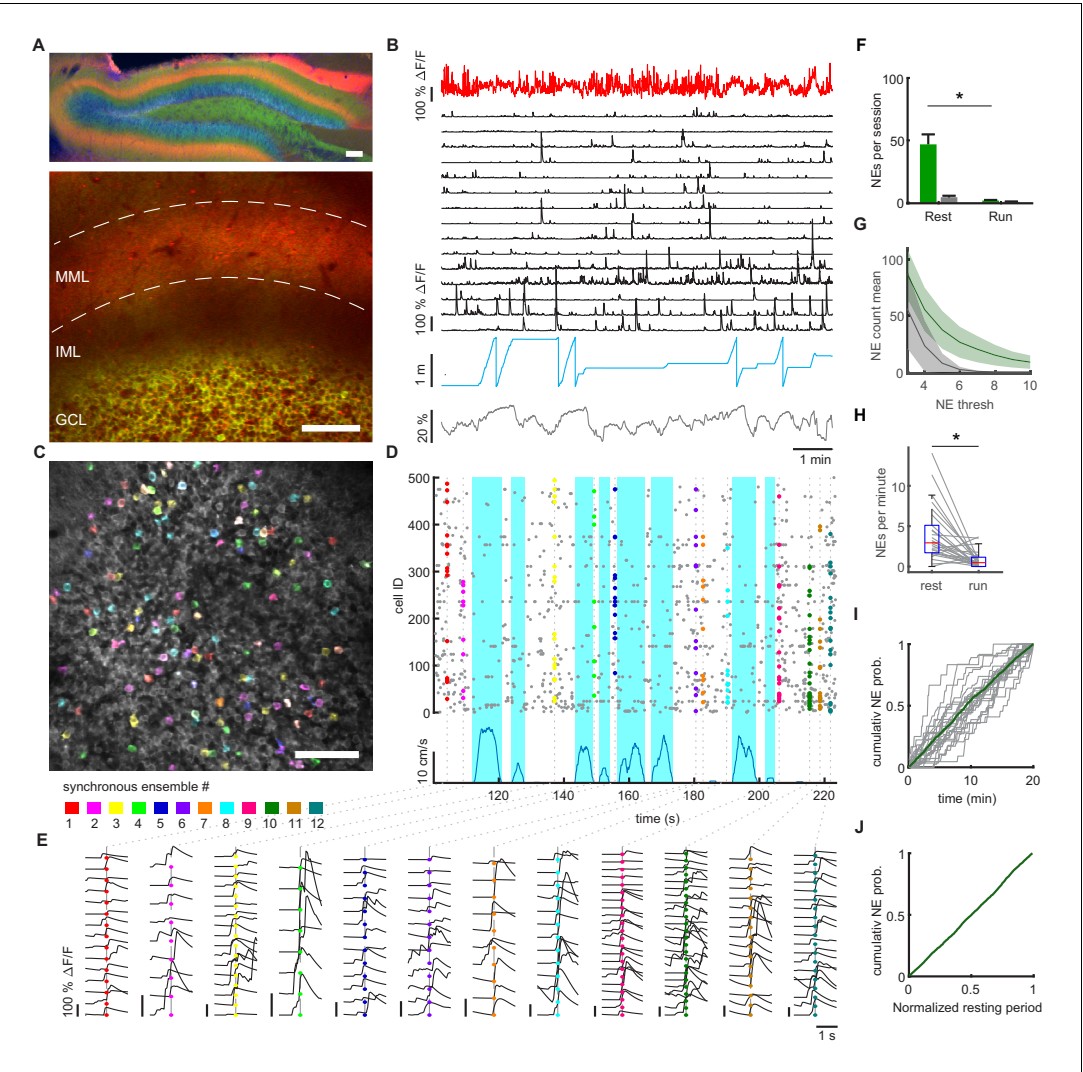

**Figure 1.** Synchronous dentate granule cell activations, 'network events', occur preferentially during immobility. (**A**) Expression of GCaMP6s in granule cells (Thy1-GCaMP mouse line, GP4.12Dkim/J). jRGECO was expressed in medial entorhinal cortex neurons using rAAV mediated gene transfer, and is visible in the middle molecular layer (MML) corresponding to the medial perforant path (MPP). Upper panel: Post hoc analysis in 70 μm fixed slice. Nuclei stained with DAPI (blue). Lower panel: Imaging plane for simultaneous recording of MPP bulk and individual GC activity. Scale bar 100 μm. (**B**) Data of representative recording session. Bulk fluorescence signal of MPP fibers (red), extracted fluorescence signals from a subset of individual GCs (black), mouse position on linear track (blue) and diameter of mouse pupil (black). (**C**) Participation of granule cells in synchronous network events. Representative field of view with highlighted simultaneously active GCs. Cells active during an individual network event are depicted in the same color. A subset of neurons is active in multiple network events, recognizable as white color. (**D**) Raster plot of network events. Dashed lines mark network events, corresponding to simultaneous activity of > four cells. Participating cells are highlighted according to color scheme from panel C. Running speed is depicted (blue) to distinguish running and resting periods. (**E**) Fluorescence transients of participating granule cells from NEs in panel D. Color scheme corresponds to panel C and D. Each column shows all respective transients of the respective synchronous ensemble. Shown is a time window of ±1 s around each NE. Vertical scale bars correspond to 100% ΔF/F. (**F**) Mean number of network events per twenty-minute recording session during running and resting. Network events occurred mainly during immobility (repeated measures ANOVA, $F_{(1,8)}$=71.80, p=2×10$^{-7}$, n = 9 animals, three sessions). Gray bars depict shuffled data for each condition (n = 9 animals). (**G**) Average number of identified network events plotted against different thresholds for the size of network events in terms of numbers of synchronously active cells (green line, shaded area depicts SEM, n = 3 mice, three sessions). Shuffled data null-distribution is created by randomly shuffling event times for every individual cell (gray line, shaded area depicts SEM). (**H**) Frequencies of network events calculated from equal time intervals for locomotion and immobility (Repeated measures ANOVA, $F_{(1,8)}$ = 117.28, p=2×10$^{-6}$). (**I**) Cumulative probability distribution of network event occurrence during the entire twenty-minute session for individual sessions (gray lines) and the pooled sessions (Green line, n = 9 mice, three sessions). (**J**) Pooled cumulative probability distribution of network event occurrence during resting periods. All resting periods of one session that were longer than 5 s were normalized to their length (n = 9 mice, three sessions).

The online version of this article includes the following video and figure supplement(s) for figure 1:

**Figure supplement 1.** Dual-color two-photon imaging in the dentate gyrus.

*Figure 1 continued on next page*

**Figure supplement 2.** Granule cell activity during locomotion on empty textured belt.
**Figure supplement 3.** Shuffling analyses demonstrating that network events do not arise by chance.
**Figure 1—video 1.** Video showing activity of granule cells and MPP, corresponding to *Figure 1A,B*.
https://elifesciences.org/articles/65786#fig1video1
**Figure 1—video 2.** Video showing network events, corresponding to *Figure 1C*.
https://elifesciences.org/articles/65786#fig1video2

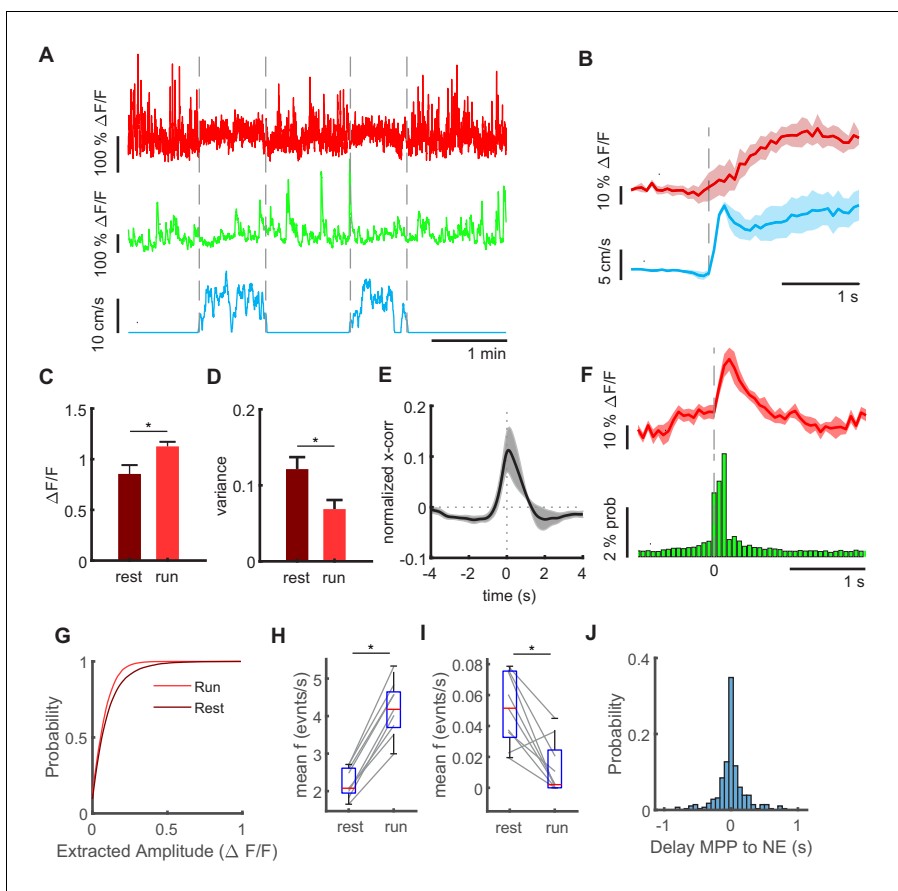

**Figure 2.** Granule cell and MPP activity during locomotion and immobility. (**A**) Mean MPP activity (red) and the sum of all granule cell activities (green) for a representative section of a recording session in an individual mouse. Dashed lines mark transition between resting and running periods (see blue line indicating running speed). (**B**) Average MPP fluorescence and running speeds, both aligned to running onsets (dashed line). Shaded areas indicate standard error (n = 4 mice, one mouse had only jRGECO expression in MPP, but no granule cell signal, three sessions per mouse). (**C**) Mean fluorescence averaged during resting (dark red) and running (light red). Asterisk indicates 5% significance threshold (repeated measures ANOVA, $F_{(1,3)}$ = 7,86, p=0.032, n = 4 mice, three sessions). (**D**) Variance of MMP bulk signal during resting (dark red) and running (light red, n = 4). Asterisk indicates results of repeated measures ANOVA, $F_{(1,3)}$ = 7,07, p=0.037, n = 4 mice, three sessions. (**E**) Cross-correlation of MPP bulk signal and summed GC signal during resting. Shaded gray area indicates standard error. (**F**) Average MPP activity (red) and probability of granule cell activity across all mice, three baseline sessions each, both aligned to the time point of network events (n = 3 mice, 2008 network events, shaded red area depicts SEM). (**G**) Amplitudes of deconvolved events in MPP bulk data during locomotion (light red) und resting (dark red). (**H**) Frequencies of all deconvolved MPP bulk events during locomotion and immobility (repeated measures ANOVA, $F_{(1,2)}$ = 255, p=3×10$^{-6}$, n = 3 mice, three sessions). (**I**) Frequencies of deconvolved events with amplitudes above two standard deviations (repeated measures ANOVA, $F_{(1,2)}$ = 27, p=2×10$^{-3}$, n = 3 mice, three sessions). (**J**) Delay of NEs to the closest identified MPP event.

n = 3 mice, three sessions per mouse, Granger causality test p<0.05). This correlation was clearly visible during network events, because aligning GC activity and MPP activity to the timepoint of network events revealed a strong coactivation of GC and MPP during network events (*Figure 2F*, n = 3 mice). During running, MPP signals did not correlate with average GC activity, which is not unexpected given the asynchronous activation of GCs during running. 8/9 sessions, Granger causality test p>0.05.

To explore in more detail how individual fluctuations in MPP bulk activity are associated with GC activity, we used a deconvolution algorithm to identify synchronous activity of MPP axons visible in the bulk MPP transients (see Materials and methods). We then quantified amplitude and frequency of these transients during locomotion and immobility. First, we found that bulk MPP events detected during immobility are on average larger than those detected during running (*Figure 2G*, Kruskal-Wallis test, n = 3 mice, three sessions per mouse, 8469 and 19,106 events during running and resting, respectively, $p=2\times10^{-44}$), in agreement with the larger variance of the MPP signal during these periods. When we examined the frequencies of all detected MPP bulk events during locomotion and immobility, we found that there were significantly more events during running (repeated measures ANOVA, $F_{(1,2)}=255$, $p=3\times10^{-6}$, n = 3 mice, three sessions, *Figure 2H*). However, large events, defined as bulk MPP events with amplitudes above two standard deviations of the mean, were significantly more frequent during resting states (repeated measures ANOVA, $F_{(1,2)}=27$, $p=2\times10^{-3}$, n = 3 mice, three sessions, *Figure 2I*). Again, this is consistent with the larger variance of the MPP signal during immobility, and likely reflects synchronized activation of MPP fibers. In line with the correlation of MPP and GC signals, there was a short temporal delay between individual bulk MPP transients and network events (*Figure 2J*).

## Dentate network events are correlated with pupil constriction

Pupil diameter is an indicator of neuronal state and arousal (*Reimer et al., 2014*; *Reimer et al., 2016*), and can be used to track changes in neuronal states during quiet wakefulness (*Reimer et al., 2014*). Of note, pupil changes have been shown to closely track the rate of occurrence of hippocampal synchronous activity, namely sharp waves in the hippocampal CA1 region (*McGinley et al., 2015*). We therefore asked if dentate network events are also associated with specific changes in pupil diameter (*Figure 3A* for example measurement of pupil diameter over multiple resting and locomotor states). As previously described (*Reimer et al., 2014*; *Reimer et al., 2016*), we found pupil constriction during immobility with dilation at locomotion onsets (*Figure 3B,C*, n = 6 mice, three sessions). Intriguingly, the average pupil diameters during network events were significantly more constricted compared to the average pupil diameters during entire periods of immobility (*Figure 3D*, repeated measures ANOVA for all three groups $F_{(2,28)}=17.17$, $p=1\times10^{-5}$, n = 6, data from three sessions each, Bonferroni post-tests: pupil diameters during locomotion vs. immobility p=0.0068, locomotion vs. network events p=0.0016, immobility vs. network events p=0.0017).

When looking at pupillary dynamics by assessing the rate of diameter change, locomotor episodes were on average associated with pupil dilation, while network events were specifically associated with pupil constriction (*Figure 3E*, repeated measures ANOVA $F_{(2,28)}=34.18$, $p=3\times10^{-8}$, n = 6 mice, data from three sessions each, Bonferroni post-tests: pupil diameters during locomotion vs. immobility p=0.0016, locomotion vs. network events $p=2.53\times10^{-5}$, immobility vs. network events p=0.00053). The latter finding was clearly illustrated by averaging pupil diameters aligned to NE times (*Figure 3F*). Together, this suggests that network events are associated with specific pupillary dynamics reflecting substates of arousal and neuronal synchronization during immobility.

## Network events are more orthogonal than expected by chance, but repetitively recruit GC sub-ensembles

We then further characterized the participation of dentate granule cells in network events. We first asked to what extent individual network events recruit orthogonal cell populations. Indeed, while individual GCs can partake in multiple network events (see *Figure 1C,D*, *Figure 1—video 2*), we also observed network events that seemed completely distinct to others. To quantify how similar network events are to one another, we computed population vectors for each network event. We then computed the cosine similarity as a measure of similarity between vectors representing individual network events (see Materials and methods). With this measure, network event pairs recruiting

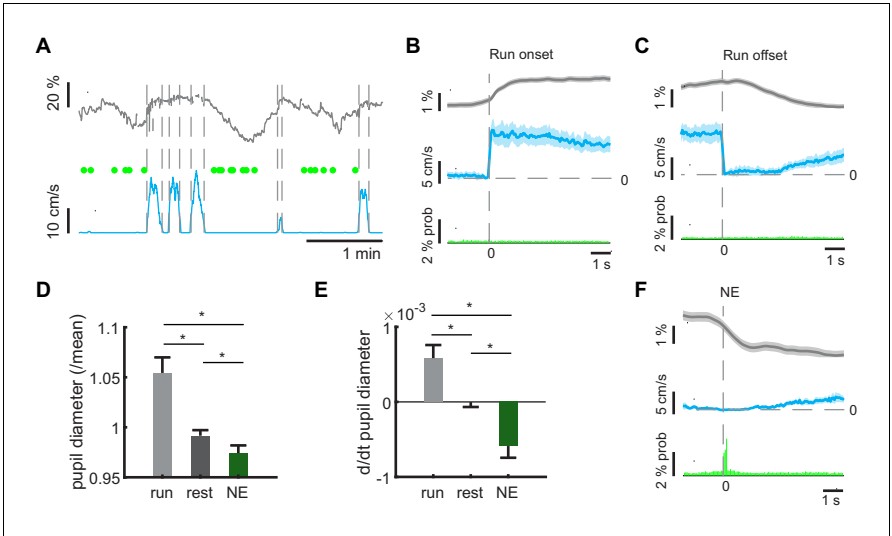

**Figure 3.** Pupil dynamics during network events. (**A**) Representative example of pupil size measurements during different locomotor states. Green dots indicate timepoints of network events. (**B, C**) Average pupil diameters (gray lines) aligned to locomotion onsets (**B**) or offsets (**C**, blue lines) reveals pupil dilation at locomotion onsets, and constriction during locomotion offset. Shaded areas indicate standard error. GC activity stays on baseline value during change of behavioral state (green). (**D**) Average pupil diameters during locomotion, during resting periods, and during network events. Asterisks indicate significant Bonferroni post-test at 5% level. (**E**) Average rate of pupil diameter change during locomotion, during resting periods, and during network events. Asterisks indicate significant Bonferroni post-test at 5% level. (**F**) Averaging pupil diameters aligned to NE times (green bars) reveals pupil constriction during NEs. Shaded areas indicate standard error. (n = 6 mice, three sessions). (n = 6 mice, three sessions).

the same set of neurons have a cosine similarity of 1, and completely orthogonal patterns exhibit a cosine similarity of 0. This analysis revealed that 38% of network event pairs were completely orthogonal to one another (*Figure 4A*, the fraction of completely orthogonal patterns corresponds to the bar with a cosine similarity of zero). Because in sparse activity patterns, orthogonality can and will arise by chance, we additionally performed a shuffling analysis to ascertain if sparse activity per se can account for the observed occurrence of orthogonal patterns. We found significantly more orthogonality than expected by chance (38 ± 4% vs. 29 ± 4% in real vs. shuffled data, respectively, see *Figure 4A* inset, n = 9 mice, three sessions, comparison to shuffled data: Wilcoxon test, p=0.0039). This is consistent with the capability to represent separate sets of information within network events.

Even though orthogonal network events were observed, we also observed a repeated activation of granule cells in multiple network events (see i.e. *Figure 1—video 2*). To examine if specific sub-ensembles of granule cells are repeatedly recruited in network events, we performed a pairwise Pearson's correlation of the activity of all cell pairs during all network events of a recording session (correlation coefficients depicted in the correlation matrix in *Figure 4B*). We then re-arranged the cells by hierarchical clustering. Clusters were combined using a standardized Euclidean distance metric and a weighted average linkage method (*Figure 4C*, more examples in *Figure 4—figure supplement 1A–H*).

This visualization reveals the existence of subgroups of cells that are strongly correlated within network event-related activity (*Figure 4C*), as previously demonstrated for activity during immobility in the CA1 region (*Malvache et al., 2016*). To more rigorously define what we considered a cluster showing correlated activity, we used a comparison to a null distribution generated by shuffling. Such approaches have been shown to outperform other approaches to define how many clusters are present in complex data (*Tibshirani et al., 2001*). We combined clusters until the mean of the cluster internal r-value reached a significance threshold, which was defined by creating a null-distribution of r-values from shuffled datasets (indicated with a vertical line in *Figure 4C*). Thus, clusters were defined quantitatively as exhibiting a mean correlation coefficient within the cluster above chance

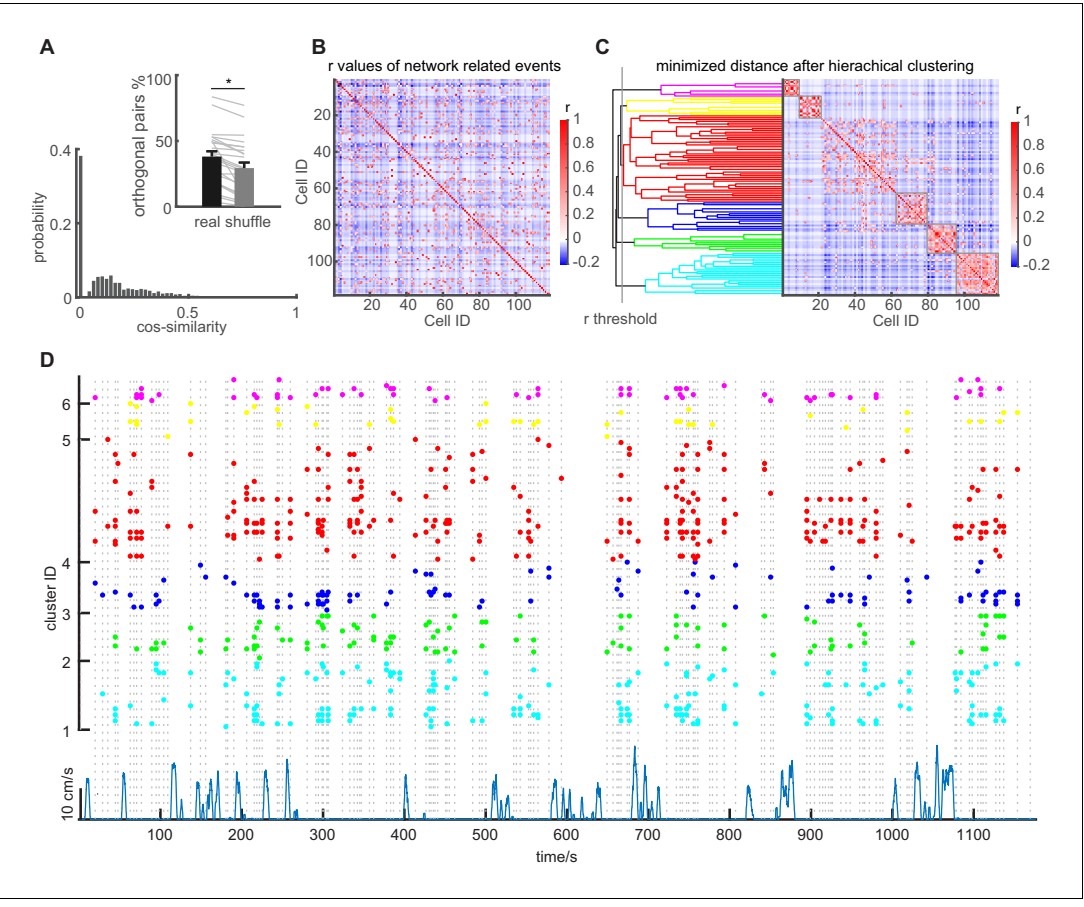

**Figure 4.** Network events are orthogonal, but repetitively recruit GC sub-ensembles. (**A**) Similarity between network events. Similarity of population vectors computed for individual network events. Comparisons were carried out between all possible pairwise combinations of vectors and quantified using cosine similarity. Inset: Mean number of orthogonal NEs for baseline sessions (Black bar) compared to shuffled data (gray bar) with SEM. Gray lines depict individual sessions. (**B**) Graphical representation of the correlation matrix using Pearson's r for all cell combinations, with values for r being color coded. Data from one representative session in an individual mouse. (**C**) Identification of clusters of correlated cells using agglomerative hierarchical clustering. Clusters were combined using a standardized Euclidean distance metric and a weighted average linkage method, until the mean of the cluster internal r-value reached a significance threshold. The 5% significance threshold was defined by creating a null-distribution of r values from randomized data sets, and is indicated for this particular experiment with a vertical line. Right panel in C depicts the reordered correlation matrix showing clusters of highly correlated cells. Only clusters whose mean intra-correlation exceeded the threshold were included in further analysis (significant clusters indicated with gray frames). (**D**) Raster plot showing the reactivation of clusters identified in panel C during multiple episodes of running and immobility. Individual dots indicate participation of individual cells. Clusters are color-coded according to the agglomerative tree. Network events are indicated by vertical dashed lines. Running episodes are indicated at the lower border with the running speed (blue).

The online version of this article includes the following figure supplement(s) for figure 4:

**Figure supplement 1.** Clustering of cells active during network events into correlated sub-ensembles via a correlation matrix.

level. Using this definition, the average cluster size was 6.7 ± 0.4 cells per cluster (n = 9 mice, three sessions). The repetitive nature of GC cluster activation during an entire session becomes clearly apparent when viewing cell activity during network events over an entire session, sorted by their participation in clusters (example shown in *Figure 4D*).

## Participation of place- and speed-coding granule cells in network events

To ask if network events carry specific spatial or locomotion-related information, we identified GCs with position-related or speed-related activity. We first identified the group of GCs that exhibited significant place coding (2.83% of n = 1415 active cells imaged in nine mice, *Figure 5A* for representative polar plots of three GCs). The place fields of place-coding GCs were distributed over the linear track (*Figure 5B*). If the fraction of place-coding cells was calculated as a fraction of only those GCs active during running, the fraction of significantly place-coding GCs was 6.09%. Secondly, we identified a fraction of GCs (0.85% of GCs, 1.83% of running-active GCs, n = 9 mice) displaying a

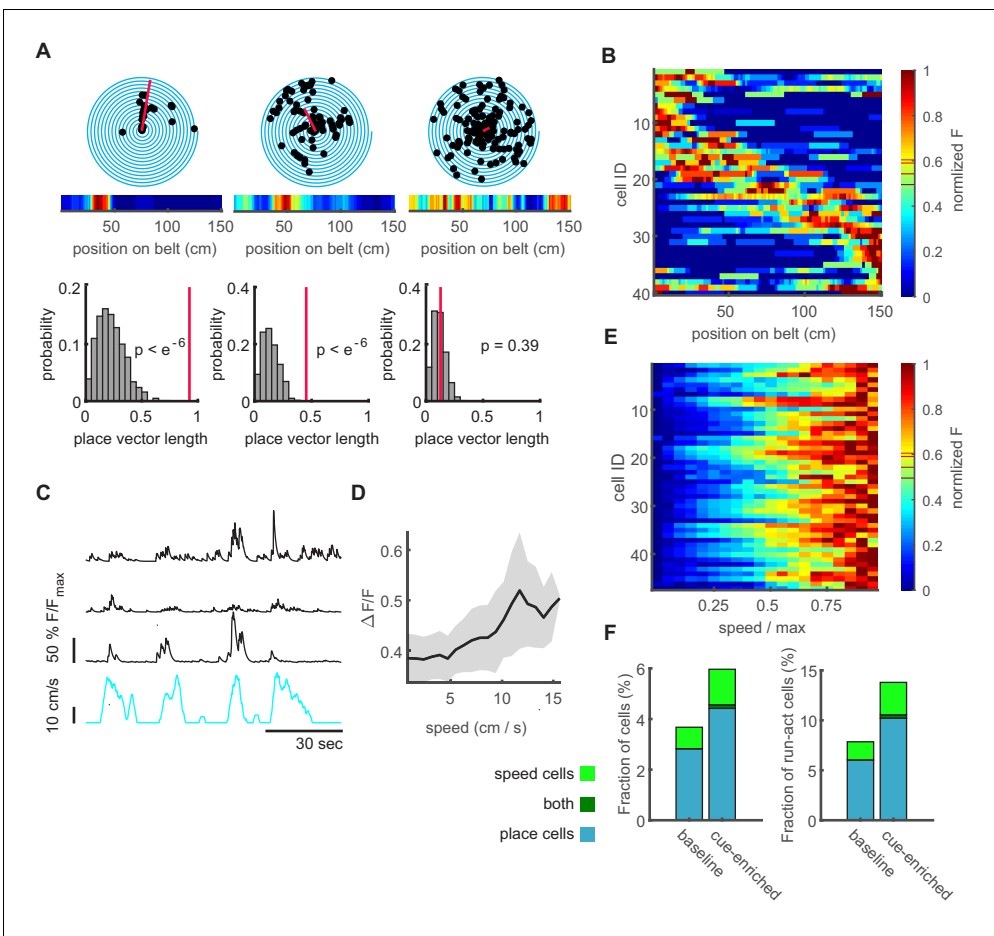

**Figure 5.** Characterization of dentate gyrus place and speed neurons. (**A B**) Place cells in the dentate gyrus. (**A**) Representative polar plots of two significantly place-coding granule cells (left, middle), and one without significant place preference (right). Place coding is depicted as spiral plot, where each 360° turn of the spiral represents a transition through the 1.5 m linear track without additional cues (baseline condition). Detected events are shown as black dots. The red line represents the place vector. The corresponding heatmap of normalized fluorescence is shown in the inset. In lower panels, the distributions for place vector lengths generated from shuffled data (see Materials and methods) are shown (gray histograms), the place vector for the individual cell is indicated by the red line. (**B**) Place field heatmaps of cells showing significant place preference. (**C**) Representative examples of three significantly speed-modulated neurons (black traces, running speed depicted in cyan). (**D**) Speed-modulated mean fluorescence signal of a representative example cell. Gray area indicates standard error. (**E**) Mean fluorescence signals of all significantly speed-modulated cells. Normalized fluorescence is color coded and running speed is normalized to every individual mouse maximum running speed. (**F**) Fractions of place and speed coding cells (cyan and green bar, respectively) normalized to all active cells (left panel) and only running active cells (right panel). Only a very small number of cells carried encoded both features (dark green bar).

The online version of this article includes the following figure supplement(s) for figure 5:

**Figure supplement 1.** Activity of granule cells and MPP inputs in cue-enriched conditions.

significant correlation of activity with running speed (*Figure 5C–E*). This is in contrast to a previous study (*Danielson et al., 2016*), but consistent with data obtained in freely moving mice (*Stefanini et al., 2020*). We also examined recordings from sessions using two other linear track environments with sensory cues placed on the textured belt. First, additional sensory cues were placed randomly on the belt (cue-enriched condition). Under these conditions, the fraction of place cells observed within the GC population increased (4.56% of GCs, 10.74% of running-active GCs), as did the proportion of speed cells (1.54% of GCs, 3.64% of running-active GCs, n = 1425 active GCs imaged in nine mice, *Figure 5F*).

Second, we tested if there is a further increase in place cells with a commonly used linear track divided into zones, each with very different spatial cues (see Materials and methods, *Figure 1—figure supplement 1G–J*). This was not the case. In these mice (n = 3), we recorded 690 GCs, of which 2.61% were place cells. As a fraction of those GCs active during running, we found 8.11% place cells. In all conditions, few cells exhibited both speed coding and place coding.

We then examined if place or speed cells are incorporated in network events, and if this participation is altered when the environment changes. Specifically, we examined the difference between the baseline linear track without additional cues and the cue-enriched condition. We chose the cue-enriched condition for further experiments and analyses because it provided sufficient spatial cues for a strong spatial representation, without introducing edges between differently cued zones on the linear track. We found that in the cue-enriched condition, dentate gyrus network events were again observed predominantly during immobility (*Figure 5—figure supplement 1D*, statistics of GC activity in *Figure 5—figure supplement 1A–C*) and were similarly related to MPP activity (*Figure 5—figure supplement 1F–J*). Increasing the cue density did not significantly alter the network event frequency (*Figure 5—figure supplements 1D*, 2.39±0.73 vs 3.63 ± 0.90 events/minute, respectively, n = 9 mice, two-way ANOVA, baseline vs. cue enriched: $F_{(1,30)}=0.71$, p=0.41, run vs. rest: $F_{(1,3)}=59.13$, p=0.001). However, the average size of individual network events, measured as the number of participating GCs, was significantly larger in the cue-enriched condition compared to the

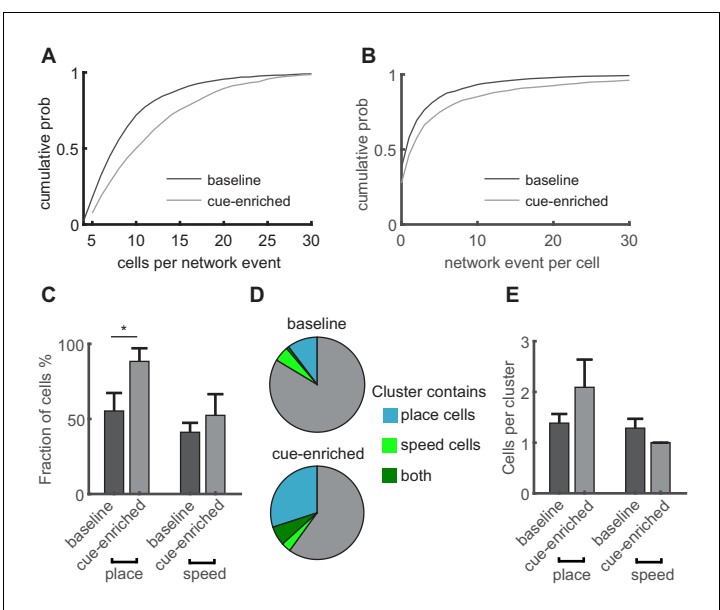

**Figure 6.** Increasing sensory cues is associated with enlargement of network events and increased incorporation of place cells. (A) Network events comprise more granule cells in cue-enriched environments. Cumulative probability of network event size (number of cells per network event) for baseline and cue enriched condition (dark and light gray lines, respectively). (B) Cumulative probability of participation in multiple network events per cell for baseline and cue enriched condition (dark and light gray lines, respectively). (C) Fraction of place and speed cells that participate in network events (total number of place/speed cells equals 100%). (D) Relation of place and speed cells to correlated cell clusters (c.f. *Figure 4*). Fraction of the total number of clusters containing place cells (cyan), speed cells (light green), or both (dark green). Gray indicates clusters containing neither place nor speed cells. (E) Mean number of place and speed cells per cluster, in baseline and cue-enriched conditions, n = 9 mice.

baseline condition (***Figure 6A***, Kruskal-Wallis test, n = 9 mice, 1313 and 1493 network events in baseline and cue-enriched condition, respectively, p=4×10$^{-41}$), with individual GCs contributing more frequently to network events in the cue-rich condition (***Figure 6B***, Kruskal-Wallis test, p=1×10$^{-40}$). Fewer orthogonal networks were observed in the cue-rich condition, but this was not significantly different to the baseline condition (not shown, Kruskal-Wallis test, n.s. p=0.49).

We then examined if the participation of place and speed cells in network events is altered in the cue-enriched compared to the baseline condition. As stated above, place cells are more commonly observed in cue-enriched sessions. However, when we calculated the fraction of all place cells that participated in network events, taking into account the total number of place cells under each condition, the probability of being incorporated in network events was increased significantly (***Figure 6C***, 55.42 vs. 88.46% of place cells in baseline vs. cue-rich conditions). This was not the case for speed cells (42.86 vs. 52.63% of speed cells in baseline vs. cue-rich conditions, n = 9 mice, chi$^2$ test regarding changes in the incorporation of place and speed cells in network events p=3×10$^{-4}$, post-test: place cells baseline vs. cue-enriched p=1×10$^{-5}$, indicated with asterisk in ***Figure 6C***, speed cells baseline vs. cue-enriched p=0.22). Thus, irrespective of the increase in the number of place cells in cue-enriched conditions, the probability of individual place cell to be integrated a network event is significantly higher. Accordingly, the proportion of synchronous events that incorporated at least one place cell increased (from 23 ± 9 to 33 ± 11%). The properties of correlated cell clusters within network events did not change (cluster size comparison, Kruskal-Wallis test, n = 9 mice, p=0.13), but significantly more of the clusters contained place cells in the cue-rich condition (***Figure 6D***, n = 9 mice, Chi$^2$ test p=0.004, post-test comparison baseline vs. cue-enriched for place cells p=0.004, speed cells p=0.6), with the number of place or speed cells per cluster remaining unchanged (***Figure 6E***).

Thus, network events are responsive to changes in the environment, and incorporate more place-coding neurons into correlated activity patterns.

## Similarity of population activity patterns during locomotion to network events during immobility

The incorporation of place and speed cells in network events, as well as the fact that changing features of the environment modifies network event size and place cell participation is consistent with the idea that animals, when immobile, represent information about the environment in synchronous, sparse network events. Testing this idea is difficult, however, given that place cells are less prevalent in the dentate gyrus compared to other hippocampal sub-regions. It has been suggested that the dentate gyrus utilizes a population code (***Stefanini et al., 2020***), meaning that even though only few cells can be rigorously classified as place cells, many more neurons may encode relevant but partial information about the environment. We used three different approaches to assess similarity between running and resting activity in the dentate gyrus at the population level. All these approaches are based on analyzing population coding separately during either locomotion or network events using Principal Component Analysis (PCA).

To obtain a first visual impression of population behavior during linear track locomotion, we plotted the neuronal state captured by the first three components (***Figure 7B,C***, ***Figure 7—figure supplement 1B–C*** for Independent Component Analysis, ICA, and Gaussian Process Factor Analysis, GPFA). We observed smooth, large trajectories with high variability reflecting movement along the linear track for some laps on the linear treadmill. Such large trajectories did not occur for every lap. We examined this unexpected phenomenon in both the baseline and cue-enriched condition (***Figure 7—figure supplement 2A,B***), as well as in the belt with three distinct zones (***Figure 7—figure supplement 2C***). In all three types of linear tracks, we found a similar, high lap-to-lap variability in the dentate gyrus population. To see how this behavior compares to the CA1 region, which is known to exhibit a reliable place code over these timeframes (***Rubin et al., 2019***), we examined CA1 neurons in mice running on a linear track with zones (n = 2 mice, 543 CA1 neurons, identical conditions to the zoned belt used for GC measurements). Here, PCA trajectories showed a much lower lap-to-lap variability and related smoothly to the position on the linear track (***Figure 7C***, ***Figure 7—figure supplement 1D–F***, for PCA, GPFA, ICA, see Materials and methods, and ***Figure 7—figure supplement 2D***). We have quantified this phenomenon across all laps in a session by plotting the weights of the first five components of the PCA across laps. In this depiction for CA1, as well as the three different versions of the linear track used for DG experiments, it is very clear that strong periodicity for

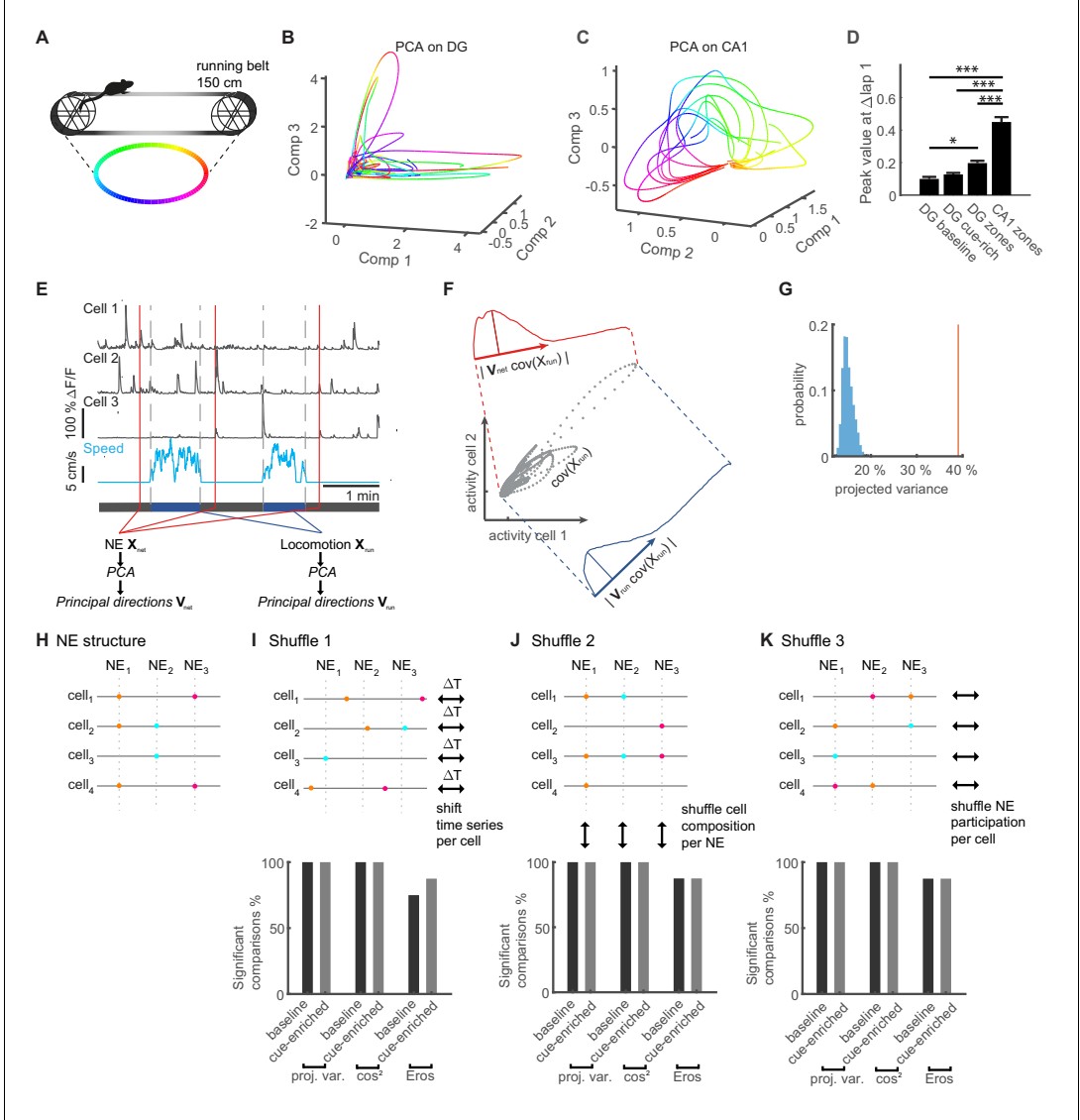

**Figure 7.** Similarity of activity patterns during network events to population patterns during locomotion. (A) Color code for position on the linear track used in panels B. (B) Trajectories during an individual representative session plotted in a three-dimensional coordinate system corresponding to the first three PCA components (Comp 1–3). (C) Trajectories calculated from CA1 $Ca^{2+}$-imaging data during an individual representative session for comparison. (D) Average peak value of weight-autocorrelations at $\Delta$lap = 1 (ANOVA, $F_{(1,3)}$ = 88.32, p=2×$10^{-30}$, * Bonferroni post-test at p<0.05, *** Bonferroni post-test at p<0.001). (E) Schematic of the procedure for comparing population activity during network events (NE) and locomotion. Population activity is represented by three cells (upper traces), recorded during running and quiet immobility (blue trace indicates speed). Time point of three network events is indicated schematically by red lines. Activity during network events (NE) was used to perform PCA, computing the transformation matrix $V_{net}$. Similarly, PCA was performed on the neuronal population activity only from running periods (speed indicated in blue, bordered by vertical gray dashed lines), to generate the transformation matrix $V_{run}$ representing the covarying activity during locomotion. (F) Schematic description of the procedure for projecting co-variances of running activity into the PCA basis of network events (or shuffled data). Gray dots show covarying activity of two representative cells during running. The blue graph denotes the projection into the locomotion PCA-space using $V_{run}$ and the width of the distribution shows the projected variance. The red graph shows the same information for the network space using $V_{net}$. (G) Individual example of shuffle analysis for a representative session. The vertical red line indicates the observed projected variance explained normalized to variance explained in the original space (50% of the overall variance). The observed variance explained is larger than the shuffled distribution (blue bars), indicating that the population activity during locomotion and network events is more similar than expected by chance (i.e. for network activity without correlations). (H) Cartoon illustrating NE structure for four cells and three synchronous events. (I) Upper panel: Cartoon illustrating the first shuffling procedure where each cells time series is shifted by a randomized time interval. Lower panel: Fraction of sessions in which comparisons of population activity were significant vs. chance level for the two cue conditions and all similarity measures (n = 8 mice, one session per condition, see *Figure 7— figure supplement 3* for comparisons to shuffled datasets for all sessions). (J) Upper panel: Cartoon illustrating the second shuffling procedure where cell IDs within each NE are randomly shuffled. This approach randomizes NE-composition while maintaining the number of cells per NE. Lower panel:

*Figure 7 continued on next page*

*Figure 7 continued*

Analogous to I. (**K**) Upper panel: Cartoon illustrating the third shuffling procedure where the NE participation is randomly shuffled for each cell. This approach randomizes NE-participation while maintaining the activity level for each cell. Lower panel: Analogous to I.

The online version of this article includes the following figure supplement(s) for figure 7:

**Figure supplement 1.** Analysis of population activity in dentate gyrus and the CA1 subfield of the hippocampus using PCA, ICA, and GPFA.

**Figure supplement 2.** PCA-based analysis of spatial representation in DG and CA1.

**Figure supplement 3.** Similarity of individual network events to population activity during running.

each round is observed in CA1, but much less so in all DG experiments (*Figure 7—figure supplement 2E–H*).

To quantify the strength of lap-periodicity (i.e. population spatial stability throughout a session) across animals, we performed an autocorrelation for all experiments in the four conditions. The autocorrelation showed large magnitude peaks at integer multiples of 1 lap for CA1, which were significantly larger than corresponding peaks for all linear track conditions in DG (examples shown in *Figure 7—figure supplement 2I–L*, averages across all mice and sessions *Figure 7—figure supplement 2M–P*, statistics *Figure 7D* ANOVA, $F_{(1,3)}=88.32$, $p=2\times10^{-30}$, * Bonferroni post-test $p<0.05$, *** Bonferroni post-test $p<0.001$).

Thus, the population behavior in DG was similar across three different types of linear track, with an episodic nature that was clearly distinct from the repetitive, stable population dynamics in CA1.

After applying PCA to locomotor states in the dentate gyrus, we then also performed a PCA analysis of population activity during network events, including the number of components explaining 50% of the variance (see Materials and methods, *Figure 7E* for schematic description). In order to compare the two sets of PCAs representing population activity during running states and network events, respectively, we first used a vector-based similarity measure. Briefly, we projected the traces recorded during locomotion into the PCA-space representing activity during network events, and tested how much of their variance was captured by them. In this analysis, similarity between both population measures would result in a large fraction of explained variance (*Figure 7F*).

To obtain the expected null distribution, we performed different types of shuffling analysis on the resting activity (see Materials and methods). In the first shuffling procedure, we shifted the entire ΔF/F time series of each cell by random time values (compare *Figure 7H,I*). Thus, all non-random activity timing between cells is destroyed and cells will no longer be synchronously active at NE timepoints. At the same time, individual cell event statistics will be maintained (i.e. inter-event-intervals). This method thus preserves intra-neuronal correlations and event frequencies, but destroys inter-neuronal correlations. The distributions from shuffled data were clearly distinct from the real data (red vertical line in *Figure 7G*, *Figure 7—figure supplement 3* for comparisons to shuffled data for all sessions). The comparisons to shuffled data were significant in all sessions, both for baseline and cue-enriched conditions (*Figure 7I*, leftmost bars in lower panel), indicating that synchronous activity is important for the similarity between locomotor related activity and network events.

We used two further similarity measures that have been used so far to quantify similarity between PCA bases. Firstly, we used a similarity factor $S_{PCA}$ as described by *Krzanowski, 1979* and the EROS similarity factor (*Yang and Shahabi, 2004*) (see Materials and methods for description), testing them against shuffled datasets in the same manner (*Figure 7I*, *Figure 7—figure supplement 3* for comparisons to shuffled data for all sessions). With these measures, significant comparisons to shuffled data were obtained with all (cosine similarity) or a majority (EROS) of sessions (*Figure 7I*, n = 8 animals, last baseline session and cue-enriched session).

This shuffling approach (*Figure 7I*), however, does not specifically test if the composition of NEs matters for the similarity between running and NE activity. We therefore implemented two additional shuffling approaches that probe the importance of NE structure by shuffling activity within NEs. In our second shuffling approach, we tested if the composition of individual NEs is important. To this end, for each individual NE, we randomly reassigned a given cells activity to a different cell. Thus, NEs have exactly the same number of active cell's, but the identity of cells active within them has been randomly changed, and the number of NEs that individual cells participate in will be altered (see schematic in *Figure 7J*, compare to panel H). This shuffling approach also revealed that

NEs are significantly more similar to locomotor related activity with all three similarity measures (*Figure 7J*, lower panel).

If morpho-functional properties in the network simply confine activity during run and rest to very specific populations of cells that are always very active, then a different type of shuffling would be required to test if this phenomenon drives similarity. We therefore added a third shuffling method, in which for each cell, we randomly reassigned its NE activity to other NEs (see schematic in *Figure 7K*, cf. panel H). Thus, how many NEs a given cell participates in is maintained. At the same time, NE interactions between specific sets of cells will be altered, although highly active cells that participate in multiple NEs will still be more likely to be co-active in shuffled NEs. If the similarity were driven by such a population of always-active cells, then this shuffling would not disrupt the similarity between running and shuffled NE activity. However, also here NE activity was more similar to running activity than shuffled data for all three similarity measures (*Figure 7K*, lower panel).

Collectively, these data show that at the population level, NEs and locomotion-related activity are more similar than expected by chance. Moreover, the two shuffling procedures described in *Figure 7J and K* suggest that the cellular composition of network events matters for this similarity.

In CA1, replay of place cell sequences has been described extensively. To ascertain the robustness of our similarity measures, we have applied them to CA1 population activity, in exactly the same manner as described in *Figure 7F*. This approach showed significant similarities between synchronous CA1 events during immobility, and activity during locomotion in 100% of the tested sessions for all three PCA-based measures (five mice, three sessions per mouse, data not shown).

## Inhibition of dentate granule cell activity during immobility disrupts pattern separation

Collectively, these data suggest that during immobility, GCs engage in structured ensemble activity that reiterates activity during running at the population level. This suggests that such activity might be important for the formation of hippocampal dependent spatial memories. The ideal test of this hypothesis would be to detect network events in freely moving animals using two-photon imaging during a memory task, and then applying closed-loop inhibition of granule cells during this task. The sparseness of granule cell activity, and the difficulties inherent in triggering closed-loop inhibition to very sparse activity patterns renders this experiment extraordinarily difficult. We therefore opted to use closed-loop inhibition of granule cells during all periods of immobility during a dentate gyrus-dependent memory task to test if dentate gyrus activity during immobility is necessary for memory formation. We used an established memory task for spatial object pattern separation (OPS, *van Goethem et al., 2018*), in which DG-dependent spatial discrimination is assessed based on the differential exploration of two objects. Briefly, animals are first exposed to two objects in defined locations during an acquisition trial (5 min) and are then re-exposed following an intermediate period, with one of the objects slightly displaced. Increased exploration of the displaced object indicates that the animal has encoded the initial location and is able to discriminate the displaced object. In preliminary experiments, we tested 4 degrees of object displacement along a vertical axis (3–12 cm, *Figure 8—figure supplement 1C*). We then determined the extent to which spatial object pattern separation was dependent on the activity of the dentate gyrus. We expressed either halorhodopsin (eNpHR, *Gradinaru et al., 2008*), or eYFP (control group) selectively in dentate GCs using Prox1-Cre mice, which efficiently inhibited GC firing (*Figure 8—figure supplement 1F–J*), and bilaterally illuminated the dentate gyrus with two implanted light fibers during the OPS task (*Figure 8—figure supplement 1A,B,D*). We found that GC activity was most important for an intermediate degree of displacement (9 cm), while maximal displacement was no longer dependent on GC activity (*Figure 8—figure supplement 1E*).

We then used this intermediate degree of displacement for the further experiments. We first inhibited GCs during locomotion only in the learning trial. As expected, inhibiting GC activity when mice actively explored the environment to be memorized led to a loss of preference for the displaced object in the subsequent recall trials (*Figure 8—figure supplement 2*).

We then used the intermediate degree of displacement in the OPS task to see if dentate gyrus activity during quiet immobility was equally required to establish a memory of object location. We bilaterally inhibited GCs during periods of quiet immobility (running speed <4 cm/s) only during the learning trial (*Figure 8A,B*). This manipulation led to a complete loss of preference for the displaced object in the subsequent recall trials (*Figure 8H* for representative sessions, analysis of

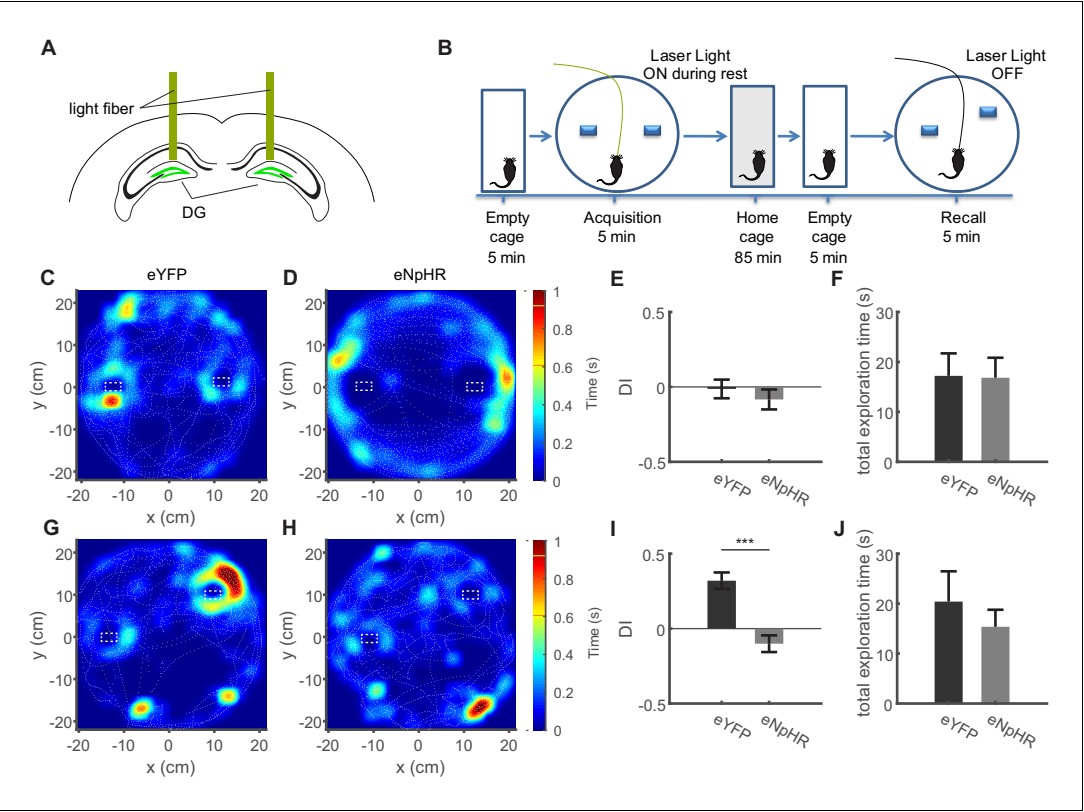

**Figure 8.** Inhibition of dentate granule cell activity during immobility prevents memory acquisition. (**A**) Schematic of the bilateral optogenetic inhibition of the dentate gyrus granule cells expressing eNpHR. (**B**) Schematic of the experimental procedure. In the acquisition phase, mice were familiarized with an arena containing two objects. Following an intermediate period of 90 min, the mice were placed in the same arena in which one object was moved slightly. (**C, D**) Representative sessions from acquisition trials in control (eYFP) mice and mice expressing eNpHR in granule cells showing the tracking of the mouse center of mass (dashed white lines), as well as normalized occupancy within the arena. (**E**) Discrimination index from the acquisition trial quantifying the specific exploration activity of the objects relative to one another (see Materials and methods), with 0 values indicating equal exploration (see Materials and methods). (**F**) Total time spent exploring the objects in the eYFP and eNpHR groups during the acquisition trial. (**G, H**) Representative sessions from recall trials depicted as shown in B, C. (**I**) Discrimination index for recall trials, showing strong preference for the displaced object in the eYFP group, but not the eNpHR group if granule cell activity was inhibited during acquisition trials only during immobility. (**J**) Total time spent exploring the objects in the eYFP and eNpHR groups during the recall trial (n = 6 animals for eNpHR group, n = 9 animals for eYFP group).

The online version of this article includes the following figure supplement(s) for figure 8:

**Figure supplement 1.** Establishing a dentate gyrus-dependent variant of the object pattern separation task.

**Figure supplement 2.** Inhibition of dentate granule cell activity during locomotion only in the acquisition trial impairs memory formation in the OPS task.

**Figure supplement 3.** Inhibition of dentate granule cell activity during immobility in the acquisition trial only in non-object locations impairs memory formation in the OPS task.

---

discrimination index in **I**, unpaired T-test with Welch's correction, n = 6 and 9 for eNpHR and eYFP respectively, $t_{(12)}$ = 5.37, p=0.0002), whereas control mice displayed a clear preference for the displaced object (*Figure 8G* for representative session). Similar results were obtained in a separate cohort of animals, where the difference in performance was measured in a paired experimental design (*Figure 8—figure supplement 2*, repeated measures ANOVA, $F_{(1,14)}$ = 54.58, p=0.0003. Bonferroni post-tests: no illumination vs. resting illumination, p=0.0026; no illumination vs. illumination during locomotion, p=0.0076; n = 5.).

Carrying out the OPS task in mice expressing eNpHR without illumination yielded discrimination indices indistinguishable from the control group (not shown). For the three groups, ANOVA revealed

a significant effect ($F_{(2,18)}$ = 8.52, p=0.003), with Bonferroni post-tests showing that inhibition of GCs significantly reduces performance vs. the two control groups (eYFP vs. eNpHR illuminated p=0.006, eYFP vs. eNpHR without illumination p>0.99, eNpHR with illumination vs. without illumination p=0.006).

Because mice are also immobile while examining the objects, we also performed a set of experiments in which light-stimulation was only carried out during immobility, but excluding a 4 cm zone surrounding the objects (*Figure 8—figure supplement 3A*). This experiment yielded similar results, with a virtually complete loss of object discrimination during the recall trial (*Figure 8—figure supplement 3B–I*). This effect was specific to acquisition. GC inhibition only during immobility in the recall trial (*Figure 8—figure supplement 3J*) elicited no significant reduction in the recognition of the displaced object (*Figure 8—figure supplement 3K–L*, n = 11 and 6 for eYFP and eNpHR groups, respectively, t-test with Welch correction n.s.). These data suggest that activity of GCs during rest is important to form memories that require discrimination of similar experiences.

## Discussion

The dentate gyrus has been implicated in pattern separation of sensory-driven activity patterns during experience but is also active during immobility and sleep. The properties of these latter forms of activity at the cellular level and the role they play in behavior are largely unknown. The application of multiphoton in-vivo Ca$^{2+}$ imaging allowed us to observe large-scale dentate gyrus dynamics at the cellular level and to detect a novel form of sparse, synchronized GC activity that occurs during immobility, termed dentate network events. These events were specifically modified by the environment, and showed higher similarity than expected by chance to population activity occurring during locomotion, indicating a sparse reiteration of locomotion-associated activity patterns.

Interestingly, network events were associated with pupil constriction and on average smaller pupil diameters when compared to entire periods of immobility, indicating that they may be associated with fluctuations in brain state during immobility, as described for visual cortex (*Reimer et al., 2014*). This finding is in agreement with the correlation of pupil constriction with the rate of hippocampal ripple oscillations during resting states (*McGinley et al., 2015*). The association of pupil constriction with synchronized activity is also very consistent with data from visual cortex, where brief episodes of pupil constriction during immobility are associated with synchronization and increased low-frequency oscillations (*Reimer et al., 2014*).

For neuronal activity during resting states to support learning or memory consolidation concerning a particular environment, one general requirement would be that there is reactivation of activity patterns induced by exploration of the relevant environment (*Davidson et al., 2009*; *Diba and Buzsáki, 2007*; *Dupret et al., 2010*; *Foster and Wilson, 2006*; *Girardeau et al., 2009*; *Skaggs and McNaughton, 1996*; *Wilson and McNaughton, 1994*). We have used three different similarity measures to show that this is the case in the dentate gyrus at the population level.

In addition to this general requirement, two specific features of resting activity are consistent with the formation of precise memories that conserve the pattern separation capabilities of the dentate gyrus. Firstly, the activity patterns, although sparse, should be capable of generating orthogonal ensembles representing different features of the environment. Secondly, the activity should repetitively recruit specific subsets of dentate GCs capable of instructing the formation of CA3 attractors via Hebbian plasticity mechanisms. Indeed, we found that activity during network events is sparse, recruiting just ~5–7% of the active GCs. Given that only ~50% of GCs are active in head-fixed animals (*Danielson et al., 2016*; *Pilz et al., 2016*), recruitment of dentate GCs during network events is much sparser than in other forms of activity occurring during immobility or sleep. For instance, the fraction of CA1 neurons recruited during sharp wave ripple mediated replay of behaviorally relevant sequences (*Davidson et al., 2009*; *Diba and Buzsáki, 2007*; *Dupret et al., 2010*; *Foster and Wilson, 2006*; *Girardeau et al., 2009*; *Malvache et al., 2016*; *Skaggs and McNaughton, 1996*; *Wilson and McNaughton, 1994*) is much higher than the recruitment of GCs in network events. One consequence of the sparseness of network events is that they are predicted to recruit highly constrained CA3 ensembles, both because of the sparse excitatory connectivity of mossy fibers in CA3, and because of the properties of the powerful inhibitory circuits in the CA3 region (*Acsády et al., 1998*; *Neubrandt et al., 2017*; *Neubrandt et al., 2018*). This has been suggested to

be important in the capability to store information in CA3, while conserving the pattern separation benefits of the dentate gyrus (*O'Reilly and McClelland, 1994*; *GoodSmith et al., 2019*).

We found that network events are structured, with subgroups of dentate GCs forming correlated sub-ensembles that are repeatedly recruited (see *Figure 4*). This finding is consistent with the idea that dentate GC activity recruits plasticity mechanisms to form sparse attractor-like representations in CA3 (*O'Reilly and McClelland, 1994*). A similar structure was also observed for awake hippocampal reactivations in the hippocampal CA1 region, and may serve similar plasticity mechanisms in downstream targets (*Malvache et al., 2016*). Thus, dentate activity during immobility may be important to instruct downstream ensembles to exhibit specific memory-related sequences. That the integrity of the dentate gyrus is important in determining behaviorally relevant firing patterns in CA3 has also been demonstrated by lesion experiments showing that activity of dentate GCs is necessary for increased SWRs and prospective goal-directed firing of CA3 neurons (*Sasaki et al., 2018*).

One interesting feature of population activity in the dentate gyrus during locomotion became apparent from our PCA analyses. We noted that population behavior in in the dentate gyrus was very dissimilar during different laps, even though animals traversed the identical belt sections. A qualitatively similar finding has been obtained in a recent publication, showing that even after extensive training in the very same environment on successive days, different sets of dentate granule cells were activated every day (*Lamothe-Molina and Franzelin, 2020*, Doi: https://doi.org/10.1101/2020.08.29.273391). The population dynamics that we observed in DG were very different from CA1, which expectedly shows a very robust association with space during repetitive laps. One interpretation of this finding is that the dentate gyrus amplifies small difference between laps, and is able to represent successive laps in a different way; this itself being a potential manifestation of the pattern separation capabilities of this structure. We note that while this is conceptually compelling, these experiments do not prove that this is the case.

If network events are important in memory processes, then inhibiting dentate gyrus activity during the entire period the animal is resting should impede the formation of dentate gyrus-dependent memories. It would be desirable to inhibit only network events to test this idea, but, due to the sparseness and high synchrony of these events, a closed loop approach to achieve this is not feasible. Thus, the behavioral experiments have to be interpreted with caution, as all resting activity is being inhibited, regardless of whether they constitute network events or not. Inhibiting only granule cells during immobility to test the effect this has on a dentate-dependent memory tasks should therefore be considered a hypothesis testing experiment, but does not provide definite proof of the relevance of network events. The OPS task requires storage of the initial object location with a high degree of precision that can be utilized later on for discriminating the translocated object. We show that optogenetically inhibiting GCs, even if this was done only during immobility remote from the explored objects, disrupted the capability to acquire such memories. This supports the idea that dentate network events may rapidly and flexibly introduce information about the environment into the hippocampal network, in the time intervals interspersed between episodes of exploration. Consistent with this view of 'real-time updating', we observed increased incorporation of spatial information via place cell integration into network events upon the first experience of a cue-rich environment. Inhibition of the dentate gyrus during the recall phase did not significantly inhibit task performance, consistent with the idea that recall of precise location information is achieved via activation of memory-related attractors in downstream CA3 and/or CA1 regions. Indeed, behavioral analyses combined with selective lesions of dentate gyrus and CA3 have also suggested an interaction between CA3 and DG in supporting encoding but not retrieval processes in a spatial learning task (*Jerman et al., 2006*). Moreover, disrupting dentate spikes via electrical stimulation has been shown to disrupt acquisition of hippocampal-dependent trace eyeblink conditioning (*Nokia et al., 2017*).

We also performed inhibition of dentate granule cells only during locomotion in the OPS task. This inhibition also prevented the formation of spatial memories. This may simply reflect that mice are not able to store the initial object location if exploratory activity is disrupted. On the other hand, it is possible that the dentate gyrus is encoding the experience of the initial session in the OPS task as a single sequence spanning rest and running. In this case, inhibition of population at any point during the entire experience could disrupt memory formation, and would not reflect a specific role of DG activity during rest.

Two caveats have to be considered in these behavioral experiments. First, while it is very likely that network events of a similar kind occur in freely moving mice during the OPS task, we have not explicitly shown this. A second caveat when using optogenetics for behavioral experiments are the known aberrant effects of some opsins. Because the design of our experiment involves closed-loop stimulation and requires inhibition with relatively fast kinetics, we had to select a fast inhibitory opsin for these experiments, with NpHR and ArchT as the most established opsins in this category. While rebound excitation effects have been described for NpHR, and not for the most prominent alternative ArchT following illumination (*Raimondo et al., 2012*), ArchT has pH-dependent effects in synaptic terminals, which lead to very powerful, action potential independent excitation of Arch-expressing terminals during illumination (*Mahn et al., 2016*). Because this could lead to aberrant excitation of hilar neurons during illumination, ArchT was not a viable alternative in our experiments. We therefore used NpHR as the most appropriate strategy, and utilized pulsed stimulation to minimize unwanted side effects. However, we acknowledge rebound excitation effects may be a potential confounding factor.

How do network events correspond to the different types of activity that have been described in the dentate gyrus during immobility or sleep, namely dentate spikes and sharp waves (*Bragin et al., 1995*; *Penttonen et al., 1997*)? During dentate spikes, granule cells are discharged anterogradely by entorhinal input, while they are activated retrogradely by the CA3-mossy cell feedback pathway during sharp waves (*Bragin et al., 1995*; *Penttonen et al., 1997*). We show that dentate network events are associated with MPP activation, but would be cautious in designating these events dentate spikes in the absence of parallel in-vivo electrophysiology, especially given recent descriptions of other subclasses of DG sharp waves (*Meier et al., 2020*).

Activity in the dentate gyrus may also be relevant for processes on more extended time scales, such as maintenance of established memories. For instance, pharmacogenetic inhibition of GCs induces loss of a hippocampal memory in trace eyeblink conditioning (*Madroñal et al., 2016*). Such longer time-scale coding may be mediated by processes that extend beyond local hippocampal computations. Along these lines, dentate gyrus activity during dentate spikes is associated with wide-spread increases in single-cell activity, gamma oscillations, and intraregional gamma coherence (*Headley et al., 2017*). It is thus possible that the precise activation patterns we observe in dentate gyrus here are part of a more distributed, organized activity occurring in immobile animals.

In summary, we described a novel form of synchronized, sparse network activity during immobility in DG that is potentially relevant to the formation of dentate gyrus-dependent spatial memories.

# Materials and methods

## Animals and procedures

All animal experiments were conducted in accordance with European (2010/63/EU) and federal law (TierSchG, TierSchVersV) on animal care and use and approved by the county of North-Rhine Westphalia (LANUV AZ 84–02.04.2015.A524, AZ 81–02.04.2019.A216). We used 9–12 weeks old Thy1-GCaMP6 mouse line (GP4.12Dkim/J) mice for imaging experiments, which express GCaMP6s in most hippocampal neurons (*Dana et al., 2014*). For optogenetic inhibition of the dentate gyrus granule cells, we used heterozygous Prox1-Cre animals (Tg(Prox1-cre)SJ39Gsat/Mmucd) obtained from MMRRC UC Davis as cryopreserved sperm and rederived in the local facility.

## Virus injections and head fixation

Thy1-GCaMP6 mice were anesthetized with a combination of fentanyl/midazolam/medetomidine (0.05/5.0/0.5 mg/kg body weight i.p.) and head-fixed in a stereotactic frame. 30 min prior to induction of anesthesia, the animals were given a subcutaneous injection of ketoprofen (5 mg/kg body weight). Eyes were covered with eye-ointment (Bepanthen, Bayer) to prevent drying and body temperature was maintained at 37°C using a regulated heating plate (TCAT-2LV, Physitemp) and a rectal thermal probe. After removal of the head hair and superficial disinfection, the scalp was removed about 1 cm² around the middle of the skull. The surface was locally anesthetized with a drop of 10% lidocaine and after 3–5 min residual soft tissue was removed from the skull bones with a scraper and 3% $H_2O_2$/NaCl solution. After complete drying, the cranial sutures were clearly visible and served as orientation for the determination of the drilling and injection sites. For virus injection, a hole was

carefully drilled through the skull with a dental drill, avoiding excessive heating and injury to the meninges. Any minor bleeding was stopped with a sterile pad. The target site was located as the joint of Parietal, Interparietal and Occipital skull plates. Subsequently, the tip of a precision syringe (cannula size 34 G) was navigated stereotactically through the burrhole (30° toward vertical sagittal plane, 1.5 mm depth from skull surface) to target the following coordinates: Anterioposterior [AP] measured from bregma ~4.6 mm; lateral [L] specified from midline ~3 mm; dorsoventral [DV] from surface of the skull ~4.2 mm. Virus particles (rAAV2/1-CaMKIIa-NES-jRGECO1a *Dana et al., 2016*) were slowly injected (total volume 250 nl, 50 nl/min) in the medial entorhinal cortex. Correct injection site in the medial entorhinal cortex was verified in all cases by confined expression of jRGECO1a in the middle molecular layer of the dentate gyrus (*Figure 1A*). To prevent reflux of the injected fluid, the cannula was retained for 5 min at the injection site. Optibond (Optibond 3FL; two component, 48% filled dental adhesive, bottle kit; Kerr; FL, USA) was then applied thinly to the skull to aid adhesion of dental cement. Subsequently, a flat custom-made head post ring was applied with the aid of dental cement (Tetric Evoflow), the borehole was closed and the surrounding skin adapted with tissue glue, also closing the borehole and adapting the surrounding skin with tissue glue. At the end of the surgery, anesthesia was terminated by i.p. injection of antagonists (naloxone/flumazenil/atipamezole, 1.2/0.5/2.5 mg/kg body weight). Postoperative analgesia was carried out over 3 days with 1 × daily ketoprofen (5 mg/kg body weight, s.c.).

## Window implantation procedure

Cranial window surgery was performed to allow imaging from the hippocampal dentate gyrus. Thirty min before induction of anesthesia, the analgesis buprenorphine was administered for analgesia (0.05 mg/kg body weight) and dexamethasone (0.1 mg/20 g body weight) was given to inhibit inflammation. Mice were anesthetized with 3–4% isoflurane in an oxygen/air mixture (25/75%) and then placed in a stereotactic frame. Eyes were covered with eye-ointment (Bepanthen, Bayer) to prevent drying and body temperature was maintained at 37°C using a regulated heating plate (TCAT-2LV, Physitemp) and a rectal thermal probe. The further anesthesia was carried out via a mask with a reduced isoflurane dose of 1–2% at a gas flow of about 0.5 l/min. A circular craniotomy (Ø 3 mm) was opened above the right hemisphere hippocampus using a dental drill. Cortical and CA1 tissue was aspirated using a blunted 27-gauge needle until the blood vessels above the dentate gyrus became visible. A custom-made cone-shaped silicon inset (Upper diameter 3 mm, lower diameter 1.5 mm, length 2.3 mm, RTV 615, Movimentive) attached to by a cover glass (Ø 5 mm, thickness 0.17 mm) was inserted and fixed with dental cement. This special window design allowed easy implantation and maintenance and minimized the amount of aspirated tissue. Further the geometry was optimal for conserving the numerical aperture of the objective (see *Figure 1—figure supplement 1D–F*). Postoperative care included analgesia by administering buprenorphine twice daily (0.05 mg/kg body weight) and ketoprofen once daily (5 mg/kg body weight s.c.) on the three consecutive days after surgery. Animals were carefully monitored twice daily on the following 3 days, and recovered from surgery within 24–48 hr, showing normal activity and no signs of pain. The preparation of CA1 imaging windows followed mainly the same protocol. Here only the cortex was aspirated until the alveus fibers above CA1 became visible. The silicon inset was shorter version (length 1.5 mm) of the one used for DG experiments.

## Two-photon calcium imaging

We used a commercially available two photon microscope (A1 MP, Nikon) equipped with a 25x long-working-distance, water-immersion objective (N.A. = 1, WD = 4 mm, XLPLN25XSVMP2, Olympus) controlled by NIS-Elements software (Nikon). GCaMP6s was excited at 940 nm using a Ti:Sapphire laser system (~60 fs laser pulse width; Chameleon Vision-S, Coherent) and a fiber laser system at 1070 nm (55 fs laser pulse width, Fidelity-2, Coherent) to excite jRGECO1a (see *Figure 1—figure supplement 1B*). Emitted photons were collected using gated GaAsP photomultipliers (H11706-40, Hamamatsu). Movies were recorded using a resonant scanning system at a frame rate of 15 Hz and duration of 20 min per movie.

## Habituation and behavior on the linear track

Experiments were performed in head fixed awake mice running on a linear track. Two weeks before the measurements, mice were habituated to the head fixation. Initially mice were placed on the treadmill without fixation for 5 min at a time. Subsequently, mice were head-fixed, but immediately removed if signs of fear or anxiety were observed. These habituation sessions lasted 5 min each and were carried out three times per day, flanked by 5 min of handling. During the following 3–5 days, sessions were extended to 10 min each. The duration of sessions used for experiments was always 20 min each. After habituation, mice ran well on the treadmill for average distances between 30 and 40 m per session (see *Figure 1—figure supplement 1H*). The treadmill we implemented was a self-constructed linear horizontal treadmill, similar to *Royer et al., 2012*. Three different belt configurations were used. In the first, no spatial cues were added to the belt beyond the texture of the belt itself (baseline condition). Three 20-min sessions were carried out for each mouse on consecutive days. In the second, cue enriched condition, the belt surface was equipped with tactile cues (see *Figure 1—figure supplement 1G*). In the zone condition the belt was divided in three zones where each zone contained unique tactile ques. Belt position and running speed were measured by modified optical computer-mouse sensors. All stimulation and acquisition processes were controlled by custom-made software written in LabView (*Source code 1*).

## Pupil diameter measurement and analysis

On the linear track, the pupil diameter was measured using a high-speed camera (Basler Pilot, Basler, Germany) at a framerate of 100 Hz. To estimate pupil diameter, a circular shape was fitted to the pupil using the LabView NI Vision toolbox (National Instruments), providing a real-time readout. Post-hoc, the pupil-diameter trace was normalized to its mean. As in a published study (*Reimer et al., 2014*), frames in which pupil diameters could not be obtained due to blinking or saccades were removed from the trace. The pupil diameter trace was filtered using a Butterworth low-pass filter at a cutoff frequency of 4 Hz. To match the time resolution of the imaging data, the pupil-trace was down-sampled to 15 Hz. Average pupil diameters were calculated for entire episodes of locomotion, entire periods of immobility, and for the single frame coincident with the peak of granule cell activity during network events.

## Data analysis, two-photon imaging

All analysis on imaging data and treadmill behavior data were conducted in MATLAB using standard toolboxes, open access toolboxes and custom written code. To remove motion artifacts, recorded movies were registered using a Lucas–Kanade model (*Greenberg and Kerr, 2009*). Individual cell locations and fluorescence traces were identified using a constrained nonnegative matrix factorization based algorithm and afterwards $Ca^{2+}$ events were identified with a constrained deconvolution algorithm (*Pnevmatikakis et al., 2016*). All components were manually inspected and only those that showed shape and size of a granular cell and at least one $Ca^{2+}$-event amplitude three standard deviations above noise level in their extracted fluorescence trace were kept. We binarized individual cell fluorescence traces by converting the onsets of detected $Ca^{2+}$ events to binary activity events. We did not observe any indication of epileptiform activity in Thy1-GCaMP6 (GP4.12Dkim/J) mice, in line with previous work (*Steinmetz et al., 2017*). On average 5–6% of the GC population was active during synchronized network events (Mean: 5,71%, Median: 5.03%, n = 1312 NEs, nine mice, three sessions).

## Analysis of MPP input signals

MPP input bulk signal was analyzed by setting a region of interest in the molecular layer. For that, a threshold of 50% maximum fluorescence was used within the field of view on the average projection of the movie. The bulk fluorescence signal trace was calculated as the average signal of the defined region of interest in each frame. The baseline for the bulk signal was defined as the low pass filtered signal of the raw trace with a cutoff frequency of 0.01 Hz using a Butterworth filter model. We used a constrained deconvolution algorithm (*Pnevmatikakis et al., 2016*) to create a proxy for the underlying activity of the bulk signal. This allowed for identification of precise onset times and normalized amplitude values of $Ca^{2+}$ events in MPP input data.

## Network activity

To define events of synchronized activity we used binarized data that marked the onset of each significant $Ca^{2+}$-event. First, we searched for events occurring simultaneously in several GCs within a moving time window of 200 ms which corresponds to $1 \pm 1$ frames in our recordings, where multiple events in one cell were counted as one. We then defined the distribution of synchronous events that could arise by chance in each individual session using three different shuffling approaches. Firstly, for every individual cell the event onset times were redistributed to random times, thereby conserving the mean event frequency per cell but destroying temporal correlations. This was done for every cell and repeated a thousand times to create a null-distribution of population behavior. To ascertain how robustly the data were different from the null distribution, we identified the number of synchronous events for different network event size thresholds (see *Figure 1—figure supplement 3A* for average values, green line real data, gray line shuffled data, see *Figure 1—figure supplement 3D–I* for six individual representative examples). The second shuffling approach shifted complete traces of onset times with respect to each other. This maintains within-cell correlations of firing (i.e. episodes of higher frequency firing), but reduces between-cell correlations. This is shown in *Figure 1—figure supplement 3B* (averages, green line real data, red line shuffled data). In the representative examples in *Figure 1—figure supplement 3D–I*, this shuffling approach is shown as red lines in the right-most panels (note that the three shuffling curves are closely superimposed). The third shuffling approach considers potential differences of individual cell activity levels during locomotion and immobility. To account for this, event onset times were randomly re-distributed only within these activity states (*Figure 1—figure supplement 3C* for averages green line real data, purple line shuffled data, *Figure 1—figure supplement 3D–I*, this shuffling method shown as purple line in right-most panels). All three shuffling methods reveal that significantly more synchrony is observed than expected by chance. We then set the minimal threshold for network events in each individual session at that number of synchronously active granule cells where less than 0.1% of events (p<0.001) could be explained by chance.

Orthogonality between pairs of network events was assessed using cosine-similarity measures. To this end, population vectors of all network events derived from binarized data were multiplied using the normalized vector-product in a pair-wise manner. To test which fraction of orthogonal pairs could be explained by chance, we generated a null distribution by randomly reassigning the cell participations to different population vectors a 1000 times. We then tested the real fraction of orthogonal pairs against the fraction derived from the shuffled data (*Source code 2*, *Source code 3*).

## Spatial tuning

To assess spatial tuning of activity in sparsely coding GCs we used spatial tuning vector analysis (*Danielson et al., 2016*). We restricted analysis to running epochs, where a running epoch was defined as an episode of movement with a minimal duration of 2.5 s above a threshold of 4 cm/s in a forward direction. The threshold of 4 cm/s was chosen in line with both literature using head-fixed mice (i.e. *Danielson et al., 2016*), as well as a very extensive literature in freely moving animals (i.e. *Kay et al., 2016*). Only cells with four or more event onsets during running epochs were included in the analysis. We used binarized data to calculate the mean of the vectors pointing in the mouse position at the times of transient onsets, weighted by the time spent in that bin. We addressed statistical significance by creating the null distribution for every spatially tuned cell. This was achieved by randomly shuffling the onset times and recalculating the spatial tuning vector 1000 times. The p value was calculated as the percentage of vector lengths arising from the shuffled distribution that was larger than the actual vector length.

## Velocity tuning

To analyze speed modulated activity of GCs, velocity values were divided in 20 evenly sized bins between 0 and the maximum velocity of the animal. We calculated the mean ΔF/F at all times the animal was running at velocities within each specific velocity bin. Putative speed cells were those granule cells that showed a Pearson's r of at least 0.9. To further exclude the possibility that correlations arise by chance, we shifted the individual ΔF/F traces with respect to the behavior randomly in the time domain 1000 times. The cell was considered a significant speed coder if the shuffle-data r-values were below the original one in at least 95% of the cases.

## Hierarchical cluster analysis

To find ensembles of correlated activity within network activity, we focused only on those granule cell $Ca^{2+}$ events that occurred within network events. We calculated the correlation matrix from binarized data using Pearson's r for all cell combinations. To identify clusters of correlated cells we used agglomerative hierarchical cluster trees. Clusters were combined using a standardized Euclidean distance metric and a weighted average linkage method. Clusters were combined until the mean of the cluster internal r-value reached a significance threshold. To define the significance threshold, we created a null-distribution of r-values from randomized data sets. Data was shuffled by randomly reassigning all network related events to different network events for every cell. This process was repeated 1000 times and the 95 percentiles of the created r-value null-distribution was used as the threshold for the clustering procedure. Only clusters in which the mean intra-cluster r-value exceeded the threshold obtained from the null distribution were considered for further analysis (see *Source code 4*).

## Principal component analysis

To perform Principal Component Analysis (PCA) and Independent Component Analysis (ICA) we used standard MATLAB procedures and calculated the maximal number of components. Gaussian Process Factor Analysis (GPFA) was conducted using a formerly described procedure and toolbox (*Yu et al., 2009*), that we adapted for $Ca^{2+}$-imaging data. Principals were calculated using singular value decomposition (SVD) of the data $\mathbf{X}$ of size $N$ by $T$, where $N$ is the number of cells, $T$ the number of recorded frames and the rows of $\mathbf{X}$ are the z-scored $F/F$ traces, decomposing the data-matrix as

$$\mathbf{X} = \mathbf{VW}$$

where $\mathbf{V}$ is an orthogonal matrix whose columns are the principal components, and $\mathbf{W}$ is a matrix of associated weights. For an analysis of population activity patterns relative to spatial location, we projected the animal position onto PCA trajectories, allowing us to identify loops in component space reflecting complete laps on the belt. Further we projected all individual component amplitudes onto the position of the mouse to detect repetitive patterns. This analysis had comparable results for PCA, as well as ICA and GPFA (*Figure 7—figure supplement 1A–C* for dentate gyrus, **D-F** for CA1).

## Analysis of spatial representation using PCA weights

After performing PCA, we quantified spatial representation within our data using the weights $\mathbf{W}_{\text{run}}$. To that end, we projected the amplitudes of $\mathbf{W}_{\text{run}}$ of the five first components onto the linear space defined by the 150 cm linear track. Spatial tuning leads to a harmonic behavior of amplitudes with respect to mouse position (see *Figure 7—figure supplement 2E–H*) and the periodicity was quantified using the normalized autocorrelation of each weight. In the individual examples (see *Figure 7—figure supplement 2I–L*) as well as averaged over animals (see *Figure 7—figure supplement 2M–P*), peaks in the autocorrelation were observed at integer multiples of rounds, in particular in CA1. To compare the strength of spatial tuning in different DG-experimental conditions as well as CA1 data, we quantified and averaged the peak values at a delay of one round (see *Figure 7D*).

## PCA-based comparison of population activity during running and immobility epochs

For further analysis, we restricted the number of components so that 50% of variance in each individual data set was explained. To compare running and network related epochs we calculated principal components $\mathbf{V}_{\text{run}}$ and $\mathbf{V}_{\text{net}}$ independently from each other so that

$$\mathbf{X}_{\text{run}} \approx \mathbf{V}_{\text{run}} \mathbf{W}_{\text{run}}, \quad \mathbf{X}_{\text{net}} \approx \mathbf{V}_{\text{net}} \mathbf{W}_{\text{net}}$$

, where $\mathbf{X}_{\text{run}}$ contains all the data from epochs of running and $\mathbf{X}_{\text{net}}$ the data from 2 s windows around all network events. To calculate the similarity between these two bases, the covariance of $\mathbf{X}_{\text{run}}$ was projected into the principal space of the network activity

$$\mathbf{S}_{\text{net} \times \text{run}} \approx \mathbf{V}_{\text{net}}^{\text{T}} \text{cov}(\mathbf{X}_{\text{run}}) \mathbf{V}_{\text{net}}$$

, where $\mathbf{S}_{\text{net}\times\text{run}}$ is the matrix of projected co-variances and $\text{trace}(\mathbf{S}_{\text{net}\times\text{run}})$ quantifies the amount of projected co-variance. This number was normalized to the total amount of covariance of locomotion activity in the locomotion principal space $\text{trace}(\mathbf{S}_{\text{run}\times\text{run}})$.

To compare our results against chance level, we used three different shuffling approaches to exclude possible mechanisms for similarities that could arise by chance. In the first procedure we used the entire traces recorded during immobility, randomly shift those with respect to each other in the time domain. This approach conserved individual activity levels and intra-neuronal correlations while creating randomized inter-neuronal correlations. We applied the original network times to these time shifted traces to create a random principal direction space $\mathbf{V}_{\text{rand}}$ and calculated the projected co-variances of $\mathbf{X}_{\text{run}}$ into the random-network space as $\text{trace}(\mathbf{S}_{\text{rand}\times\text{run}})$. For shuffling approaches two and three, we shuffled within the matrix of concatenated NEs – in other words, only the 2 s around NEs rather than the entire time-series used in shuffle one. For the second approach, we tested if the composition of individual NEs is important. To this end, for each individual NE, we randomly reassigned activity of a given cells activity to a different cell. Thus, NEs have exactly the same number of active cells, but the identity of cells active within them has been randomly changed, and the number of NEs that individual cells participate in will be altered. In the third shuffling method, we tested whether similarity could be driven by the activity level of individual cells within NEs. Therefore, we randomly reassigned each cells' NE activity to other NEs. This approach maintains the number of NEs a given cell participates and randomizes the interactions between specific sets of cells. All procedures were repeated 1000 times and the p-value was calculated as the percentage of random projections that exceeded the initial value.

Additionally, we used two alternative approaches to quantify similarity between the PCA bases. First a similarity factor $\mathbf{S}_{\text{PCA}}$ as described by *Krzanowski, 1979*.

$$\mathbf{S}_{\text{PCA}} = \text{trace}\left(\mathbf{V}_{\text{net}}^T \mathbf{V}_{\text{run}} \mathbf{V}_{\text{run}}^T \mathbf{V}_{\text{net}}\right) = \sum_{i=1}^{k} \cos^2 \theta_i$$

, where $\theta_i$ is the angle between the ith principal directions of $\mathbf{V}_{\text{run}}$ and $\mathbf{V}_{\text{net}}$. Further the Eros similarity factor as described in *Yang and Shahabi, 2004* was used:

$$\text{Eros} = \sum_{i=1}^{k} w_i \left|\cos \theta_i\right|$$

where $w_i$ is a weighting factor. All measures delivered comparable results as compared to shuffled data. All different procedures of similarity calculation and shuffling available with this paper (see *Source code 5*).

## In vitro patch-clamp experiments

Acute slices were prepared from mice expressing NpHR-eYFP in GCs. NpHR expression was induced by rAAV mediated gene transfer (rAAV2/1-DOI-eNpHR3-eYFP) into Prox1-Cre mice (see below for virus injection protocols). >2 weeks after virus injection, animals were deeply anesthetized with Isoflurane (Abbott Laboratories, Abbot Park, USA) and decapitated. The head was instantaneously submerged in ice-cold carbogen saturated artificial cerebrospinal fluid (containing in mM: NaCl, 60; sucrose, 100; KCl, 2.5; NaH$_2$PO$_4$, 1.25; NaHCO$_3$, 26; CaCl$_2$, 1; MgCl$_2$, 5; glucose, 20) and the brain removed. Horizontal 350-µm-thick sections were cut with a vibratome (VT1200 S, Leica, Wetzlar, Germany). Slices were incubated at 35°C for 20–40 min and then stored in normal ACSF (containing in mM: NaCl, 125; KCl, 3.5; NaH$_2$PO$_4$, 1.25; NaHCO$_3$, 26; CaCl$_2$, 2.0; MgCl$_2$, 2.0; glucose, 15) at room temperature. Recordings were performed in a submerged recording chamber at 33–35°C under constant superfusion with carbogen saturated ACSF (3 ml/min). Visually identified GCs were recorded in current clamp using a low chloride intracellular solution (containing in mM: K-gluconate, 140; 4-(2-hydroxyethyl)−1-piperazineethanesulfonic acid (HEPES-acid), 5; ethylene glycol tetraacetic acid (EGTA), 0.16; MgCl$_2$, 0.5; sodium phosphocreatine, 5) and a Multiclamp 700B and Digidata 1322A (Molecular Devices). Illumination (~560 nm,~1 mW) was achieved through the Objective. Action potential frequencies were calculated using the smallest current injection yielding at least four action potentials.

## Light fiber implantation for behavioral experiments

Mice were injected with buprenorphine (0.05 mg/kg BW) 30 min before inducing anesthesia using 3.5% isoflurane for induction and 1–1.5% for maintenance. Mice were placed in a stereotactic frame (Kopf Instruments) and the scalp opened with surgical scissors after disinfecting it with iodine solution and applying local anesthetic (10% lidocaine). The skull was thoroughly cleaned using 2% $H_2O_2$, covered with a thin layer of two-component dental adhesive (Optibond) and the surrounding wound sealed with tissue glue (Vetbond). Small craniotomies were performed above the target sites and 500 nl virus suspension (rAAV2/1-Ef1a-DOI-NLS-eYFP for controls, rAAV2/1-EF1a-DOI-eNpHR3-eYFP for experimental group) was bilaterally injected using a 34 G syringe (Nanofill Syringe, World Precision Instruments, Inc) at a speed of 50 nl/min. After each injection, the syringe was kept in place for 5 min to ensure permeation of the virus into the parenchyma. Coordinates for viral injections into dorsal dentate gyrus were: (from bregma): AP: −2.3; ML: -/+ 1.6 and DV: 2.5 mm. Afterwards, fiber optic cannulae of 200 µm diameter (NA: 0.39, CFMLC12, Thorlabs) were bilaterally implanted at (from bregma): AP: −1.7; ML: -/+ 1.35 and DV: 1.7 and fixed to the skull with a layer of flowable opaque composite (Tetric Evoflow) topped by multiple layers of dental cement (Paladur, Heraeus). Finally, antibiotic cream (Refobacin, Almirall) was applied to the wound and the animals received ketoprofen (5 mg/kg BW) s. c. Analgesia was applied post-surgery by injecting ketoprofen (5 mg/kg BW) s. c. after 24, 48, and 72 hr. All mice recovered for at least 3 weeks after surgery before the start of behavioral experiments.

## Behavioral experiments

To test the effect of optogenetic inhibition of granule cells, 23 heterozygous Prox1-Cre animals (5 male, 18 female) between the age of 4 and 9 months were used. Males were single caged and females were group-caged (2–4 individuals per cage) in standard mouse cages under an inverted light-dark cycle with lights on at eight pm and ad libitum access to food and water. Prior to experiments, animals were randomly assigned to the control or experimental group. All experiments were conducted during the dark phase of the animals. Prior to experiments, mice were handled by the experimenter for at least 5 days (at least 5 min/day). On experimental days, animals were transported in their home cages from the holding facility to the experimental room and left to habituate for at least 45 min. All experiments were performed under dim light conditions of around 20 Lux. Animals were also habituated to the procedure of photostimulation by attaching a dual patch cord for around 10 min to the bilaterally implanted light fibers and letting them run in their home cage on multiple days.

For optogenetic stimulation, 561 nm laser light (OBIS/LS FP, Coherent, Santa Clara, CA) was delivered bilaterally into the implanted optical fibers using a dual patch cord (NA: 0.37, Doric lenses, Quebec, Canada) and a rotary joint (FRJ, Doric lenses, Quebec, Canada) located above the behavioral test arena. The laser power was set to 5 mW at the tip of the fiber probes.

We used a pulsed laser light at 20 Hz with a 50% duty cycle instead of continuous illumination to keep the light-induced heat effect minimal in the brain. Previous studies have shown that continuous light delivery to brain tissues can cause a temperature increase of up to 2°C (*Owen et al., 2019*). It has been reported that changes in temperature can alter neuronal physiology of rodents (*Stujenske et al., 2015*) and birds (*Long and Fee, 2008*). We simulated the light-induced heat effect using the model developed by *Stujenske et al., 2015*. We first examined the continuous light photostimulation of 561 nm light with 5 mW output power. The temperature increase reached a steady-state and was found to be 0.7°C in 60 s. For the pulsed light stimulation used in this study, the temperature change did not exceed 0.3°C in 60 s (see *Figure 8—figure supplement 1K–M*). Both pulsed and continuous stimulation resulted in efficient silencing of granule cell activity (*Figure 8—figure supplement 1F–J*). The average illumination times were not significantly different between eNpHR and eYFP control groups in any of the experiments. To apply photostimulation only during immobility, we used a closed loop system employing EthoVision 8.5 that opened an optical shutter (SH05, Thorlabs) in the light path of the laser via a TTL pulse only when the tracking software detected that the body center of the animal was moving less than 5 cm/s and closed the shutter if the speed exceeded 5 cm/s over a period of 0.5 s. Behavior was recorded using EthoVision 8.5 software (Noldus, Netherlands) and an IR camera with a frame rate of 24 Hz. Videos were stored on a computer for offline analysis. To apply photostimulation only during mobility, we reversed the

parameters and let the shutter open when the speed of the animal exceeded 5 cm/s and closed the shutter at speeds of below 5 cm/s for more than 0.5 s.

## Object pattern separation task

We used the object location memory test as a test for spatial learning ability. We used a circular arena (diameter: 45 cm) made of red Plexiglas with 40 cm high walls as described in *van Goethem et al., 2018*. Two pairs of almost identical building blocks (4 × 3×7 cm) made of either plastic or metal were used as objects. All building blocks were topped with a little plastic cone to prevent animals from climbing onto the objects. To habituate animals to the arena and the experimental procedure, they were taken out of their home cage and placed into an empty cage only with bedding for 5 min in order to increase their exploration time. After connecting the mice to the photostimulation apparatus via the light fiber, they were placed into the empty arena for 10 min with no laser light. On test days, mice were again first placed into a new empty cage for 5 min, then connected to the photostimulation apparatus and placed into the arena for 5 min. The arena contained two identical objects placed in the middle of the arena with an inter-object distance of 18 cm. The animals were allowed to freely explore the arena and the objects. After 5 min, the animals were transferred into their home cage for 85 min. For the recall trial, everything was done identically to the previous acquisition trial, but one of the two previously encountered objects was displaced. Experiments for individual displacement configuration were repeated in some cases up to three times with an interval of 2 days and the discrimination indices averaged. The following variants of this task regarding object displacement were performed using a first batch of up to 18 mice (3 males, 15 females). In a first set of experiments, variable displacements (3, 6, 9, and 12 cm) were used, with inhibition of granule cells carried out during the entire acquisition and recall trial (schematic in *Figure 8—figure supplement 1A–C*). In a second set of experiments a fixed, intermediate degree of displacement (9 cm, position three in *Figure 8—figure supplement 1C*) was used, and inhibition of granule cells was carried out only during quiet immobility using a closed-loop system (see above) in either only the acquisition trials (*Figure 8*) or only the recall trials (*Figure 8—figure supplement 3*). In a final set of experiments with those mice, we applied photostimulation in the acquisition trials only when the mouse was in quiet immobility and the nose point of the mouse at least 4 cm away from the objects (*Figure 8—figure supplement 3*). Lastly, with a separate batch of mice consisting of 2 males and three females, all bilaterally injected into the dentate gyrus with eNpHR-eYFP, we performed the pattern separation task with an object displacement of 9 cm either without illumination, with illumination at speeds below 5 cm/s or at speeds above 5 cm/s, both times in the acquisition trials (*Figure 8—figure supplement 2*). After each mouse, the arena and the objects were cleaned with 70% ethanol. An experienced observer blinded to the experimental group of the animals manually scored the time the animals explored the displaced and the non-displaced object by sniffing with the snout in very close proximity to the objects and their head oriented toward them. We calculated the discrimination index based on the following formula: (time exploring displaced object – time exploring the non-displaced object) / (time exploring displaced object + time exploring the non-displaced object). Trials in which the total exploration time in an acquisition or recall trial was lower than 4 s were excluded from further analysis. Trajectory maps and occupancy plots were generated using custom-written MATLAB Scripts.

## Histology and microscopy

To verify successful viral transduction and window position, animals were deeply anesthetized with ketamine (80 mg/kg body weight) and xylazine (15 mg/kg body weight). After confirming a sufficient depth of anesthesia, mice were heart-perfused with cold phosphate buffered saline (PBS) followed by 4% formalin in PBS. Animals were decapitated and the brain removed and stored in 4% formalin in PBS solution. Fifty to 70 μm thick coronal slices of the hippocampus were cut on a vibratome (Leica). For nuclear staining, brain slices were kept for 10 min in a 1:1000 DAPI solution at room temperature. Brain slices were mounted and the red, green, and blue channel successively imaged under an epi fluorescence or spinning disc microscope (Visitron VisiScope). In optogenetic inhibition experiments, post-hoc microscopy was used to confirm successful expression of NpHR-eYFP. Of the animals used, three control animals showed no eYFP expression and one experimental animal lacked NpHR-eYFP expression. The animal lacking NpHR expression was excluded from the study. Control

animals lacking eYFP expression were assumed to lack modulation of granule cell activity by illumination and were pooled with eYFP expressing control animals.

Expression of NpHR was highly selective to the dentate gyrus, as described previously (*Braganza et al., 2020*; *Truman et al., 2012*), and as reported in the Gensat project (http://www.gensat.org/ShowMMRRCStock.jsp?mmrrc_id=MMRRC:036632). Functional evidence has excluded recombination in hilar interneurons in this mouse line (*Braganza et al., 2020*). Crossing the Prox1-Cre mouse line used in the present study with a mouse leading to Cre-dependent expression of ChR2 showed a lack of monosynaptic inhibitory responses, as would be expected if ChR2 were also present in hilar interneurons.

## Acknowledgements

We are very grateful to David Greenberg, Jason Kerr, and Damian Wallace for technical help and advice, as well as the supply of analysis algorithms. We gratefully acknowledge the support of Jonathan Ewell in editing the manuscript, and Antoine Madar for helpful comments on an earlier manuscript draft. We acknowledge the support of the Imaging Core Facility of the Bonn Technology Campus Life Sciences. The work was supported by the SFB 1089, Project C04 to HB, the Research Group FOR2715, the Research Priority Program SPP Computational Connectomics and EXC 2151 under Germanys Excellence Strategy of the Deutsche Forschungsgemeinschaft (DFG, German Research Foundation) to HB and to JHM (EXC 2064/1 PN 390727645), support of the Humboldt Foundation PSI to KG, and support of the Volkswagen Foundation to LAE. ANH was supported by the IMPRS Brain and Behavior.

## Additional information

### Funding

| Funder | Grant reference number | Author |
|---|---|---|
| Deutsche Forschungsgemeinschaft | SFB 1089 Project C04 | Heinz Beck |
| Deutsche Forschungsgemeinschaft | EXC 2064/1 PN 390727645 | Jakob H Macke Heinz Beck |
| Alexander von Humboldt-Stiftung | PSI | Kurtulus Golcuk |
| Volkswagen Foundation | | Laura A Ewell |
| Deutsche Forschungsgemeinschaft | EXC 2151 | Heinz Beck |
| IMPRS Brain and Behavior | | André N Haubrich |

The funders had no role in study design, data collection and interpretation, or the decision to submit the work for publication.

### Author contributions

Martin Pofahl, Conceptualization, Software, Formal analysis, Supervision, Investigation, Visualization, Methodology, Writing - original draft, Writing - review and editing; Negar Nikbakht, Investigation, Methodology; André N Haubrich, Formal analysis, Investigation; Theresa Nguyen, Nicola Masala, Fabian Distler, Investigation; Oliver Braganza, Supervision, Investigation, Visualization; Jakob H Macke, Software, Methodology; Laura A Ewell, Conceptualization, Writing - original draft, Writing - review and editing; Kurtulus Golcuk, Software, Supervision, Visualization; Heinz Beck, Conceptualization, Resources, Supervision, Funding acquisition, Writing - original draft, Project administration, Writing - review and editing

### Author ORCIDs

Martin Pofahl https://orcid.org/0000-0002-9473-6195
André N Haubrich http://orcid.org/0000-0001-7895-6203

Oliver Braganza (iD) http://orcid.org/0000-0001-8508-1070
Jakob H Macke (iD) http://orcid.org/0000-0001-5154-8912
Heinz Beck (iD) https://orcid.org/0000-0002-8961-998X

## Ethics

Animal experimentation: All animal experiments were conducted in accordance with European (2010/63/EU) and federal law (TierSchG, TierSchVersV) on animal care and use and approved by the county of North-Rhine Westphalia (LANUV AZ 84-02.04.2015.A524, AZ 81-02.04.2019.A216).

## Decision letter and Author response

Decision letter https://doi.org/10.7554/eLife.65786.sa1
Author response https://doi.org/10.7554/eLife.65786.sa2

## Additional files

### Supplementary files

• Source code 1. Behavior apparatus LabView program.

• Source code 2. Network detection MATLAB code.

• Source code 3. Network cluster structure MATLAB code.

• Source code 4. Network shuffle analysis MATLAB code.

• Source code 5. PCA similarity MATLAB code.

• Supplementary file 1. Spread Sheet containing statistical test results for present data with respective figure numbers.

• Transparent reporting form

### Data availability

Binarized imaging traces of all cells from all experiment sessions are available on Dryad. https://doi.org/10.5061/dryad.mkkwh70z6.

The following dataset was generated:

| Author(s) | Year | Dataset title | Dataset URL | Database and Identifier |
|---|---|---|---|---|
| Pofahl M | 2020 | Synchronous activity patterns in the dentate gyrus during immobility | http://dx.doi.org/10.5061/dryad.mkkwh70z6 | Dryad Digital Repository, 10.5061/dryad.mkkwh70z6 |

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
