## [Decision Letter]

**Acceptance summary:**

This study used two-photon calcium imaging of the dentate gyrus in mice and found synchronous network events in the dentate gyrus during periods of immobility that share some similarity with activity recorded during movement. The link between this network activity and memory will be interesting to explore in future studies. The paper will be of interest to neurophysiologists studying the cellular/network mechanisms of hippocampal-dependent memory formation.

**Decision letter after peer review:**

[Editors’ note: the authors submitted for reconsideration following the decision after peer review. What follows is the decision letter after the first round of review.]

Thank you for submitting your work entitled "Dentate gyrus population activity during immobility supports formation of precise memories" for consideration by *eLife*. Your article has been reviewed by a Senior Editor, a Reviewing Editor, and three reviewers. The following individual involved in review of your submission has agreed to reveal their identity: Jerome Epsztein (Reviewer #1).

Our decision has been reached after consultation between the reviewers. Based on these discussions and the individual reviews below, we regret to inform you that your work will not be considered further for publication in *eLife*.

During the consultation session, reviewers agreed that the major conclusions of the paper are not justified without additional experiments. In other words, reviewers felt that the paper falls short of providing convincing results to support the major conclusions. A major issue raised by all reviewers is the disconnect between the imaging results and the freely moving experiment. First, reviewers agreed that there is an important control experiment missing in the freely moving experiment (inhibit only during locomotion). Second, if this control experiment is completed, then there is still a missing link between the DG network events found during imaging to the inhibition of learning during the freely moving experiments. Other effects of DG inhibition could have produced this deficit in learning. As it stands, it is unclear whether the network events occur during the pattern separation test, and so the optogenetic inactivation experiments do not show that the network events are involved in the acquisition of the task. Yet, this is not explicitly stated in the manuscript in its current form.

Another concern raised was the high variability of population activity between different laps in the DG compared to CA1 population activity recorded in the same paradigm. It is unclear what this means in terms of spatial coding and memory in the dentate gyrus during this task. The question was also raised about how this variability could affect data analysis illustrated in Figure 4, which is meant to support an important claim of the paper that activity during synchronous events "recapitulates" activity during movement.

Reviewers agreed that the topic is interesting. We would be willing to reconsider a majorly revised manuscript in the future should you find a way and should you choose to address these major concerns. We recognize that many labs are unable to collect new data at this time due to the pandemic, but reviewers still wanted to give this option to convey their enthusiasm for the questions. However, all agreed that a soft rejection is appropriate at this point (rather than Revise and Resubmit) because it's possible that the major conclusions of the paper may change based on the outcome of control experiments necessary to support the major conclusions.

Reviewer #1:

Granule cells of the dentate gyrus are numerous and discharge very sparsely despite receiving intense synaptic inputs from the entorhinal cortex. This has led to the hypothesis that the dentate gyrus would work as a filter/classifier for entorhinal inputs allowing segregation of similar contexts representations onto distinct cell assemblies. Dentate granule cells are active both during locomotion during which their firing is spatial modulated (place cells) and during immobility periods during which they are synchronously co-activated. These co-activations coincide with the recording of large fluctuations in local field potential called dentate spikes in the dentate gyrus and sharp-waves in CA1. In this paper the authors used dual-color bi-photon in vivo calcium imaging in head fixed mice walking on a treadmill to characterize the synchronous activities recorded during immobility in the dentate gyrus and compare them with activities recorded during movement. Mice were running either on an un-cued treadmill (with textures) or a cued-treadmill (with more salient tactile sensory cues). During immobility, they observe co-activation of dentate granule cells with on average 6 cells per synchronous events. These synchronous activities are temporally correlated with increased activity of axonal fibers from the medial entorhinal cortex and correspond to the reactivation of some dentate granule cells active during periods of movement including place cells (average 1.5- 2 place cells/synchronous event) and cells that code for speed (average 1.2). A number of pairs of synchronous events are orthogonal. The authors then use PCA to compare activity during synchronous events and movement. They report a high lap-by-lap variability of dentate granule cell activity (notably compared to CA1 pyramidal cell activity) during movement and more similarity between single synchronous event activity during immobility and population activity during movement compared to a shuffled distribution. Finally, they observed that inhibiting dentate granule cell activity during immobility impairs both learning and recall of an object spatial discrimination task.

1) The behavioral task used in this article is described only briefly. According to the method section animals are not rewarded. In the absence of reward is there any sensory cues structuring the task? The first belt has textures only, but the number, spatial arrangement and type of sensory information provided by these textures are not described. If the sensory information provided by textures is reduced, this may explain why few place cells are observed when the animal navigates in this environment and possibly the high lap by lap variability of their discharge (Bourboulou et al., 2019). The number of spatially modulated cells appears to be greater at the beginnings and ends of the environment. Were there specific sensory cues at these locations? How could animal know that they completed a single lap around the belt instead of travelling an infinite environment? Was there a specific reason for choosing an impoverished environment? Even when more salient tactile cues are added (cued environment) they seemed to be randomly interspaced and provide poor spatially relevant information.

2) MPP fibers seem much more active than granule cells. Could you quantify this activity for comparison (frequency and amplitude of transients)? Although synchronous activities in the granule cell network seems to be associated with an increase in MPP fiber fluorescence, there also seems to be large increases in MPP fiber activity not associated with synchronous DG activities. Can the authors quantify the proportion of increased MPP activity associated with synchronous events?

3) Could the authors discuss the lack of causality between MPP activity and granule cells' activity during movement that seems counter-intuitive at first glance, especially in an environment impoverished in sensory cues?

4) What are the respective durations of movement and immobility periods and what is the influence of this proportion on the probability of observing synchronous events during movement?

5) To what extent do the synchronous events described in this study correspond to the co-activation of granule cells observed during the dentate spike or sharp wave ripples?

6) The authors find intriguing that the synchronous activities are associated with pupil constrictions but this is also the case for the synchronous activities associated with sharp-wave ripples (McGingley et al., 2015, Figure 1).

7) The authors state that "a large fraction of events pairs were orthogonal to each other" but what is the exact fraction and how to decipher whether it is large or not?

8) Concerning the coding of place cells, what is the proportion of place cells with small and large place fields? What is the proportion of synchronous events that incorporate at least one place cell and/or one speed cell? Were place cells with a sharp place field more likely to be reactivated? What is the total number of cells imaged in this study? If 300 cells were imaged and the place cells represent 2% of the recorded cells only 6 place cells would be recorded? however at least 40 cells are shown in Figure 3.

9) In the environment enriched with tactile cues the proportion of place cells but not speed cells increases. Does this increase alone explains the higher proportion of cells per network events or were additional non-spatially modulated cells also included? This could reflect a global increase in network excitability. In general it would be interesting to know the proportion of active and silent DGCs during movement that are active during synchronous events.

10) How do authors explain the high lap to lap variability of PCA trajectories in the dentate gyrus when compared to CA1? Is it due to the low proportion of place cells in the DG or to the use of uncued treadmill? This high variability complicates comparisons between network synchronous activity and population activity during movement. Indeed, if a network event is similar to population activity during a given lap it will be different from population activity during subsequent laps. We would like to have more precisions on the selection of movement periods and synchronous activities to perform these comparisons. Did the authors compare the activity between a period of movement and the synchronous activity on a lap by lap basis?

11) Population activity during locomotion is more similar to network events than expected by chance but this is unsurprising given that the network architecture morpho-functional properties of dentate granule cells etc… will impose constrains on observed neural dynamics. A useful comparison would be with CA1 where synchronous network events that replay place cells sequences have been thoroughly described. What is the measure of similarity between synchronous events and population activity in the CA1 region? How does the similarity index in DG compares to that in CA1?

Reviewer #2:

In this manuscript, Pofahl et al., found synchronized network events in the dentate gyrus during periods of immobility in mice. These synchronized network events occurred in largely separate groups of granule cells and were correlated with increased excitation coming from the MPP. About half of the cells participating in these synchronized events encoded the position or the speed of the mouse during mobility. Increasing the cues along the track increased the number of cells participating in each network event and increase the fraction of place cells in each network event. Using dimensionality reduction methods, the authors found that a significant amount of the variance of running activity is captured by the space represented by the network events. Separately, using optogenetics the authors inhibit the dentate gyrus of mice during the learning of a pattern separation task and show inactivating dentate gyrus during immobility (away from the objects of interest) can impair acquisition of the pattern separation task. Overall, the manuscript is well-written, and the analysis is done to a high standard. I found the synchronized events in DG, and their relation to activity patterns during locomotion, to be particularly interesting.

1) There is no strong connection between the belt (linear track) head fixed task and the object recognition freely moving task. First, in the head-fixed task the authors found synchronized events during immobility that appear orthogonal to each other, but the mice were not engaged in any pattern separation task and so there is no way to deduce the relevance of this phenomenon to pattern separation. Second, using a pattern separation task in freely moving mice, the authors showed that inactivation of granule cells in the dentate gyrus during immobility can impair pattern separation, but they did not show that the synchronized events (found during the head fixed task) occur during this pattern separation task. The most significant connection between the tasks is the occurrence of rest periods in both. It is reasonable to assume that the network events occur during the pattern separation task, but the authors have no proof for this. I strongly suggest that the authors explicitly address this issue in their manuscript and simply state that it is unclear whether the network events occur during the pattern separation test, but assuming that they do occur, then their optogenetic inactivation experiments imply that the network events are involved in the acquisition of the task.

Along these same lines, the last phrase of this statement in the Discussion does not appear correct and should be revised: "These events were specifically modified by the environment, showed higher similarity than expected by chance to population activity occurring during locomotion, and were important for formation of memories requiring pattern separation."

2) In the pattern separation task, it is unclear if ANY inhibition of DG during acquisition can impair learning/memory/preference, or if it is only inhibition during immobility. It seems to me that the following control experiment is missing: inhibit only during locomotion (the same amount of time as during their immobile inhibition experiment).

Reviewer #3:

In Pofhal et al., the authors utilize 2-photon calcium imaging of dentate granule cells and MEC input in head-fixed, awake mice to show ensembles of neurons active during immobility are important for the encoding of spatially related information. They find ensembles of neurons that fire during immobility, that these ensembles are largely orthogonal, with a small subpopulation that are shared across ensembles, that network events contain place cells and speed cells, and the ensembles of neurons incorporated into network events is greater in cue rich environments. They go on to show that inhibition of DG GCs during periods of immobility (movement less than 4cm/s) during acquisition of an object place task impairs subsequent retrieval. This paper is potentially interesting and understanding the population-level encoding properties of the DG are certainly of interest to the field. However, the manuscript, as presented, does not provide a strong case that these network events are meaningful. There is no indication what these network events represent, and if they are related to anything significant in terms of dentate physiology or behavior. In addition, I don't see how links can be made between the imaging data and the behavioral data. This and other comments below:

1) Figure 1B and 1D should show the same snippet of data in order to be convinced of the fact that these events only occur during immobility. For instance it could be that during movement you simply don't see them. Figure 1D should show all events, not only the synchronous ones, (maybe highlight which cells the algorithm has delineated into a synchronous ensemble) to justify the need to focus on the synchronous ones that happen during immobility.

2) The chance level for synchronous events is very generous, as it does not take into account differences in overall activity during network events. The authors should try shuffling to check for synchronous events using time bins of equal activity levels; rather than choosing from a random time bin, time bins should be chosen from those of similar activity level of the population. In other words one needs to control for mean population activity. This would increase confidence that the patterns of synchronous events in the real data still more frequent than chance. Alternatively, a null hypothesis could be generated by simulating a group of neurons with an underlying, varying overall network activity, and make each cell fire independently at random times given that underlying network activity. You will have some neurons that fire together just by chance. So is this chance still lower than what they observe? While the network events may still be real, these would be more rigorous tests.

3) Clustering analysis: This should be assessed with silhouette scores, not mean-r only. In addition " we focused only on those granule cell Ca^2+^ events that occurred within network events", why only during network events? In addition, how was the number of clusters chosen, was there any cross-validation performed?

4) If cells are tuned so much to speed (Figure 2E) and have almost 0 activity during immobility (as in the heat map), what activity are activity is being analyzed in the network event? In other words, it's a bit confusing to consider how speed cells are involved in immobility. In addition, only ~50% of place and speed cells are recruited in network activity, does it mean that they are recruited randomly, by chance?

5) In Figure 3, is the enhancement of place and speed cells, and increased network activity due to more neural activity in the enriched environment (Figure 5—figure supplement 1A)? This possibility should be excluded.

6) I found Figure 4 confusing and difficult to interpret. They say in the abstract that they "recapitulate patterns evoked in self-motion". It’s not clear to me where this is shown. Maybe I don't understand this figure, but I was expecting they would do PCA during locomotion and compare to PCA during immobility. I’m also confused by Figure 4E, what is being conveyed here? What is the variance explained in their PCAs? Do the first 3 component explain a good fraction of the variance?

7) When introducing the behavior, the authors state "Collectively, these data suggest that during immobility, GCs engage in structured ensemble activity that reiterates activity during running at the population level. This suggests that such activity might be important for the formation of hippocampal dependent spatial memories." However, the optogenetic experiment doesn't really test this, as the inhibition is impacting a multitude of things outside of these network effects. Also, the authors should plot total immobile time (i.e. total time of DG inhibition) for each mouse and compare to the discrimination index (do the mice that were more immobile, and thus got more inhibition show worse discrimination?). In addition, a more rigorous control would be inhibition during mobility to show that there is any specificity to the immobility period.

8) They state that "We did not observe any indication of epileptiform activity in Thy1-GCaMP6 (GP4.12Dkim/J) mice, as already reported," however this should be shown, specifically in the mice with the 3mm-1.5mm cone implant above dentate gyrus. It would be reassuring to the reader to show that these network events are not due to abnormal activity from the preparation.

[Editors’ note: further revisions were suggested prior to acceptance, as described below.]

Thank you for submitting your article "Dentate gyrus population activity during immobility supports formation of precise memories" for consideration by *eLife*. Your article has been reviewed by Laura Colgin as the Senior Editor, a Reviewing Editor, and three reviewers. The following individual involved in review of your submission has agreed to reveal their identity: Jerome Epsztein (Reviewer #1).

The reviewers have discussed the reviews with one another and the Senior Editor has drafted this decision to help you prepare a revised submission.

We would like to draw your attention to changes in our policy on revisions we have made in response to COVID-19 (https://elifesciences.org/articles/57162). Specifically, when editors judge that a submitted work as a whole belongs in *eLife* but that some conclusions require a modest amount of additional new data, we are asking that the manuscript be revised to limit claims to those supported by data in hand, or to explicitly state that the relevant conclusions require additional supporting data.

Summary:

This study used two-photon calcium imaging of the dentate gyrus in mice and found synchronous network events in the dentate gyrus during periods of immobility that share some similarity with activity recorded during movement. The synchronized events and their relation to activity patterns during locomotion were interesting and novel. The authors also found that silencing dentate gyrus activity during immobility impaired learning in an object location discrimination task performed by freely moving mice. The authors conclude that synchronized events in the dentate gyrus during immobility are involved in memory acquisition, but this conclusion did not have strong support, as Figure 8—figure supplement 2 shows that inactivation of the dentate gyrus during locomotion also impairs learning. This suggests that any inhibition of the dentate gyrus during acquisition of the task can impair learning, and thus conclusions about the specificity of the events during resting are not supported. Still, reviewers found the paper to be interesting because of the synchronized events in dentate gyrus and their relationship to activity patterns during locomotion. Reviewers agreed that the link between this network activity and memory would be interesting to explore in future studies. The paper will be of interest to neurophysiologists studying the cellular/network mechanisms of hippocampal dependent memory formation.

Essential revisions:

1) In the first round of review, the primary concern was the link between network events during immobility and the behavior in the discrimination task. In the revised manuscript, the strong claims of the paper are still not strongly supported by the results shown, and the conclusions of the manuscript should be toned down accordingly and caveats clearly described and discussed. In the revised manuscript, the control experiment has now been provided in Figure 8—figure supplement 1, and the reviewers appreciate this. However, Figure 8—figure supplement 1 shows that inactivation of DG during locomotion in the object place discrimination task also impairs learning. This information is important since it suggests that ANY inhibition of DG during acquisition of the task can impair learning and so it is not clear if there is anything unique about the DG activity during resting for acquisition. Perhaps DG is encoding the experience as a single sequence spanning rest and running, and breaking this sequence at any point could then impact learning. This is an important caveat that should be added as a counterpoint to statements such as the following in the Abstract "Using optogenetic inhibition during immobility, we show that granule cell activity during immobility is required to form dentate gyrus-dependent spatial memories." Also, in the Discussion, the authors should include a statement such as the following "however, the same inhibition of DG during locomotion in the OPS task also prevents the formation of spatial memories, making it unclear if DG activity during rest plays a role in acquisition that is unique or different from the activity during locomotion". In general, any claims about specificity of network events during immobility to memory operations, or misleading language, should be revised and toned down. The authors should describe the data as they stand: the synchronized events during immobility may be involved in learning; however, any specificity remains unknown and untested here.

2) The huge inter-trial lap by lap variability of population activity in DG during running could be due to the lack of salient sensory cues at fixed locations in the belt or reward. In their response to the previous round of review, the authors suggest that activity in DG is structured by laps because a fixed sensory cue (a change in belt width) is present. But this information is absent from the Materials and methods section, and it is difficult to determine if animals take into account this sensory cue. Authors should find some indication either in their data and in the analysis of behavior that this is indeed the case. The high inter trial similarity of PCA trajectories in CA1 and the data showing that part of the belt is overrepresented seems to suggest that this is the case but this point should be discussed and clearly stated in the manuscript to justify the lap by lap analysis.

3) For the orthogonality of network events, authors should indicate the percentage of orthogonal network events that are likely to occur by chance to give readers an idea of the percentage of "true" orthogonal events.

4) The variability of PCA trajectories in DG should be quantified and compared to CA1. The reviewers would like to have actual numbers in the text for comparison. Also, PCA trajectories in CA1 should be illustrated in Figure 7 for comparison.

5) Reviewers would be more convinced about the specificity of the reinstated activity during synchronous network events by a comparison with random network activity during immobility but outside synchronous network events or a shuffling procedure which preserves inter-neuronal correlations such as procedure number 2 in Figure 1—figure supplement 3.

6) Reviewers recommend deciphering clusters with a different shuffling procedure maybe leading to a higher threshold because some of these clusters do not look convincing in Figure 4 and Figure 4—figure supplement 1.

Reviewer #1:

Granule cells of the dentate gyrus are numerous and discharge very sparsely despite receiving intense synaptic inputs from the entorhinal cortex. Dentate granule cells are active both during locomotion during which their firing is spatially modulated (place cells) and during immobility periods during which they are synchronously co-activated during large fluctuations in local field potential called dentate spikes in the dentate gyrus and sharp-waves in CA1. In this paper the authors used dual-color bi-photon in vivo calcium imaging in head fixed mice walking on a treadmill to characterize at the cellular level the synchronous activities recorded during immobility in the dentate gyrus and compare them with activities recorded during movement. Mice were running either on an un-cued treadmill (with textures) or a cued-treadmill (with more salient tactile sensory cues). During immobility, they observe co-activation of dentate granule during synchronous network events. These synchronous activities are temporally correlated with increased activity of axonal fibers from the medial entorhinal cortex and correspond to the reactivation of some dentate granule cells active during periods of movement including place cells and speed cells. A number of pairs of synchronous events are orthogonal but synchronous events also reactivate similar clusters of DGCs. The authors then use PCA to compare activity during synchronous events and movement. They report a high lap by lap variability of dentate granule cell population activity (notably compared to CA1 pyramidal cell activity) during movement on the belt and some similarity between synchronous event activity during immobility and population activity during movement. Finally, they observed that inhibiting dentate granule cell activity during immobility impairs performance in an object spatial discrimination task.

Synchronous activation of principal cells during awake immobility and sleep has been well described in the CA1 area of the hippocampus and linked to hippocampal dependent spatial memory formation. Much less is known for synchronous activities of dentate granule cells of the dentate gyrus during immobility. Given the sparse activity of these cells two photon calcium imaging seems an appropriate approach to address this question. The study provides interesting characterization of synchronous network activity in the dentate gyrus during immobility. Interpretation of the results is however limited by the high lap by lap heterogeneity of dentate granule cells population activity in the behavioral paradigm used. Furthermore the link between these synchronous network events and memory during an object spatial discrimination task deserves to be strengthened in future studies.

1) In the behavioral paradigm used, head-fixed animals are running on belts which can be devoid of any sensory cue or with randomly interspaced cues. Furthermore no reward is provided at a fixed location/distance on the belt. In these conditions it is unclear whether animals get a sense of the dimension of the environment and notably whether they completed single laps around the belt instead of travelling an infinite environment. This point is important because several analyses are performed lap by lap notably the characterization of dentate gyrus place cells in Figure 3 and comparisons between PCA trajectories in Figure 7.

2) Authors use a shuffle distribution to compare population activity during running periods and population activity during immobility-associated synchronous network events and observed that similarity is higher than expected by chance, but this is unsurprising given that the network architecture morpho-functional properties of dentate granule cells etc… will impose constrains on observed neural dynamics. Furthermore, the shuffle method used eliminates inter-neuronal correlations. To decipher the specificity of this similarity, a more convincing comparison would be with activity recorded during immobility but outside network events. Alternatively authors could use a shuffling method that preserves inter-neuronal correlations.

3) Trajectories in PCA space show high inter-trial variability in the dentate gyrus unlike what is observed in CA1. This could result from the lower proportion of spatially modulated cells in the dentate gyrus compared to CA1. This heterogeneity raises a question about the specificity of reinstated activity during synchronous network events.

4) Authors report preferred reactivations of identical cell clusters during synchronous network events. However, while some clusters illustrated in Figure 4C look convincing (like the red cells' cluster) others appear not very convincing (like the orange cells' one in Figure 4 C). Authors could use a more stringent shuffling procedure to strengthen cluster detection.

Reviewer #2:

In this manuscript, Pofahl et al. found synchronized network events in the dentate gyrus during periods of immobility in mice. These synchronized network events occurred in largely separate groups of granule cells and were correlated with increased excitation coming from the MPP. About half of the cells participating in these synchronized events encoded the position or the speed of the mouse during mobility. Increasing the cues along the track increased the number of cells participating in each network event and increase the fraction of place cells in each network event. Using dimensionality reduction methods, the authors found that a significant amount of the variance of running activity is captured by the space represented by the network events. Separately, using optogenetics the authors inhibit the dentate gyrus of mice during the learning of a pattern separation task and show inactivating dentate gyrus during immobility (away from the objects of interest) can impair acquisition of the pattern separation task. However, inactivation of DG during locomotion in the same task also impairs learning. This is an important caveat since it suggests that any inhibition of DG during acquisition of the task might impair learning and so it is not clear if there is anything unique about the DG activity during resting for acquisition. Overall, the manuscript is well-written, and the analysis is done to a high standard. The synchronized events in DG and their relation to activity patterns during locomotion are particularly interesting.

Reviewer #3:

The authors have nicely addressed my technical concerns in the first round of review. However, a direct link between the network events that occur during immobility and learning is not shown, nor is specificity for immobility shown, as inhibition during mobility also produces deficits in learning.

---

## [Author Response]

[Editors’ note: the authors resubmitted a revised version of the paper for consideration. What follows is the authors’ response to the first round of review.]

Reviewer #1:1) The behavioral task used in this article is described only briefly. According to the method section animals are not rewarded. In the absence of reward is there any sensory cues structuring the task? The first belt has textures only, but the number, spatial arrangement and type of sensory information provided by these textures are not described. If the sensory information provided by textures is reduced, this may explain why few place cells are observed when the animal navigates in this environment and possibly the high lap by lap variability of their discharge (Bourboulou et al., 2019). The number of spatially modulated cells appears to be greater at the beginnings and ends of the environment. Were there specific sensory cues at these locations? How could animal know that they completed a single lap around the belt instead of travelling an infinite environment?

We have added a more detailed description of the linear track. In brief, the animals were indeed not rewarded, as in previous imaging studies showing structured population activity in CA1 (Malvache et al., 2016). However, even the belt devoid of explicitly placed local spatial cues was not completely without spatial information, as the cut ends of the linear track belt were connected via a tape on the underside, imposing a different flexibility of the belt. We note, however, that if the fraction of place-coding cells was calculated as a fraction of only those GCs active during running (as commonly done in other publications, see (Danielson et al., 2016)), it was higher (6.09% instead of 2.83%, Results). In the cue-enriched condition, these fractions were 4.56% of all active GCs, 10.7% of running-active GCs.

For the revised manuscript, we have now performed additional experiments with a linear track divided into zones, each with very different spatial cues, as also commonly used in many studies. In these mice (n=3), we have recorded 690 GCs, of which 2.61% were place cells. As a fraction of those GCs active during running, we found 8.11% place cells, and thus no increase in place cell fraction over the cue-enriched condition. This information has been added to the manuscript Results, Figure 1—figure supplement 1).

We realize that normalizing to the total number of detected GCs may have been confusing, as most studies normalize to the cells active during running. We have tried to be clearer about this in the revised version of the manuscript, and give both fractions in the revised manuscript initially. However, since our manuscript is strongly focused on GCs active during rest, we believe that it is important to give the fraction of place cells as a fraction of total GCs detected.

Was there a specific reason for choosing an impoverished environment? Even when more salient tactile cues are added (cued environment) they seemed to be randomly interspaced and provide poor spatially relevant information.

We agree with the reviewer that it might have been better to add tactile cues with more spatially structured differences. Our thinking originally was that we would like to have spatial information equally distributed along the belt, avoiding edges between patches of completely different tactile cues. Since mice are very good at discriminating even slight differences in the tactile structure of their environment, we thought that the random arrangement of spatial cues would still provide enough spatial information. As stated above, our fraction of place-coding cells calculated as fraction of those GCs active during running was reasonable high, and did not increase substantially when using a more commonly used variation of the linear track with a segmented belt. We would therefore argue that there is sufficient spatial information in our linear track to permit us to draw conclusions about how this spatial information is represented in the network events.

2) MPP fibers seem much more active than granule cells. Could you quantify this activity for comparison (frequency and amplitude of transients)? Although synchronous activities in the granule cell network seems to be associated with an increase in MPP fiber fluorescence, there also seems to be large increases in MPP fiber activity not associated with synchronous DG activities. Can the authors quantify the proportion of increased MPP activity associated with synchronous events?

We would like to note at the outset that we have not analyzed single MPP fibers, but have recorded bulk activity of all fibers within the medial molecular layer. This is because at the depth of imaging, we could not reliably resolve single axons. Thus, in this manuscript we are interpreting the MPP signal as a summed activity of many fibers. This makes it straightforward to look at average slower changes in MPP activity during changes in locomotion.

Indeed, as we showed in the initial submission, the average MPP activity is increased during locomotion, consistent with increased sensory input (see Figure 2C in the revised manuscript). Superimposed on this steady-state activity were larger synchronous events, which we found were particularly prominent during immobility. We had previously shown that this results in a larger variance of the bulk MPP signal during immobility (see Figure 2D in the revised manuscript).

However, it is true that there are notable peaks in the bulk MPP signal, most likely representing synchronous activity of MPP fibers. Following the reviewer’s suggestion, we now additionally use a constrained foopsi deconvolution algorithm to identify individual bulk MPP transients (see Materials and methods and [Pnevmatikakis et al., 2016]). We have then, as requested, quantified amplitude and frequency of these transients during locomotion and immobility. First, we find that bulk MPP events detected during immobility are on average larger than those detected during rest (see Figure 2G in the revised manuscript). When we plotted the frequencies of all detected MPP bulk events during locomotion and immobility, we found that there were significantly more events during running (repeated measures ANOVA, F_(1,2)_ = 255, p = 3*10^-6^, n = 3 mice, 3 sessions, see Figure 2H). However, large events, defined as bulk MPP events with amplitudes above two standard deviations of the mean, were significantly more frequent during resting states (repeated measures ANOVA, F_(1,2)_ = 27, p = 2*10^-3^, n = 3 mice, 3 sessions, see Figure 2I). Regarding association of MPP events with network events, there was a close temporal association between both events (see Figure 2J).

These data thus indicate that (i) MPP fibers are more active during running than during immobility, as expected, (ii) during immobility, however, large bulk MPP events indicative of synchronous activity are more common and (iii) NEs are closely associated temporally with MPP events. As this has expanded the MPP dataset, we have condensed all data on MPP activity and GC-MPP coactivity into one figure, presented as Figure 2 in the revised manuscript.

The reviewer is right in noting that the number of detected bulk MPP events is larger than the number of NEs that we see. However, this to some extent is expected, as we are recording from a large fraction of the MEC axons in the particular hippocampal segment we are imaging, which also target GCs outside our field of view. Additionally, we are recording only from a small fraction of GCs compared to the total fraction of GCs. Thus, we are also recording synchronous MEC activity that is perhaps correlated with active GC ensembles outside (or not completely inside) our imaging field of view.

3) Could the authors discuss the lack of causality between MPP activity and granule cells' activity during movement that seems counter-intuitive at first glance, especially in an environment impoverished in sensory cues?

We are happy to discuss this. We do not believe that our results suggest that there is no relationship between MPP activity and granule cell activity during movement – rather the activity structure and the techniques that we are using are not ideal for finding such a relationship during movement. The state dependent differences we observe in the bulk signal are not only a read out for the amount of input but also for the synchrony of the imaged fiber bundle. The increase of bulk-fluorescence during locomotion reflects an overall higher asynchronous MPP-input (Figure 2B, C). This is supported by our deconvolution analysis that found significantly more small amplitude events during locomotion compared to immobility (Figure 2G, H), Overall dense asynchronous activity makes it difficult to identify direct correlations between input and output (i.e. it would only be expected that a selection of peaks drives any given active GC), especially when activity levels of input and output differ. However, during immobility we observe a sparser but more synchronous input signal. This is reflected by distinct fluorescence peaks (Figure 2A), a higher overall variance (Figure 2C) and more detected high amplitude events (Figure 2I). The significant cross-correlation analysis reveals that the synchronous input events are correlated to the network events that we observed in GC ensemble. This said, we also think that a very tight correlation between MPP activity and GC activity might not be expected across all states because of the strong feed-forward inhibition at the MPP– to DG synapse (Ewell and Jones, 2010). We have tried to make this clearer in the revised manuscript (Results).

4) What are the respective durations of movement and immobility periods and what is the influence of this proportion on the probability of observing synchronous events during movement?

This point is well taken. We now additionally report total durations of movement periods for the different linear tracks in Figure 1—figure supplement 1 of the revised manuscript.

In addition, we now present the frequencies of NEs during locomotion and immobility, using equal times for locomotion and immobility periods. NE frequencies were significantly higher during immobility (repeated measures ANOVA, F_(1,8)_=117, p=2*10^-6^, n=9 mice, 3 sessions). These analyses were added to the Results section of the revised manuscript.

5) To what extent do the synchronous events described in this study correspond to the co-activation of granule cells observed during the dentate spike or sharp wave ripples?

We had addressed this in the Discussion section of our initial submission. In the absence of co-implanted electrodes, we cannot finally address this issue. However, we had discussed that during dentate spikes, granule cells are discharged anterogradely by entorhinal input, while they are activated retrogradely by the CA3‐mossy cell feedback pathway during sharp waves (Bragin et al., 1995; Penttonen et al., 1997). We do show that dentate network events are associated with MPP activation, however, we would be cautious in designating these events dentate spikes in the absence of parallel in-vivo electrophysiology. We have explicitly stated this in the Discussion.

6) The authors find intriguing that the synchronous activities are associated with pupil constrictions but this is also the case for the synchronous activities associated with sharp-wave ripples (McGingley et al., 2015, Figure 1).

We are very grateful for this comment. We had missed this interesting paper, and have now referenced it in our manuscript (discussion). We feel that it strengthens the general point that increased synchrony in both hippocampal and neocortex is associated with pupil constriction, well in line with our findings.

7) The authors state that "a large fraction of events pairs were orthogonal to each other" but what is the exact fraction and how to decipher whether it is large or not?

We had compared network events using cosine similarity as a measure of similarity between vectors representing individual network events. In this measure, identical patterns have a cosine similarity of 1, and completely orthogonal patterns would exhibit a cosine similarity of 0. Thus, the data we had shown in Figure 1G (Figure 4A of the revised manuscript. Specifically, the bar with a cos similarity of zero) shows that 36% of network events are orthogonal to one another. We now explicitly refer to this in the manuscript to make this clear (Results).

Because in sparse activity patterns, orthogonality can and will arise by chance, we had additionally performed a shuffling analysis to ascertain if sparse activity per se can account for the fraction of orthogonal patterns. We found significantly more orthogonality than expected by chance (n=9 mice, 3 sessions, comparison to shuffled data Wilcoxon test, p=0.0039). We presume that the reviewer is referring to the fact that the number of completely orthogonal patterns that we report does not consider that a certain number of orthogonal events would occur at a certain chance level. We agree that this makes it difficult to state that the orthogonality is “large” and now simply state that it is larger than expected by chance.

8) Concerning the coding of place cells, what is the proportion of place cells with small and large place fields?

We realized following this reviewer comment that we had designated our examples of place coding neurons shown in Figure 2 as “narrow” and “broad”. It was not our intent to state that these are two categories of place cells, but simply to illustrate that place coding is not uniform even within the place cell population. We apologize for being misleading and have corrected this.

Nonetheless, we have calculated the size of place fields according to a Gaussian fit of the average discharge frequency over space. We also quantified the precision of place coding using the vector lengths described in the original Figure 2 (now Figure 5), with longer vector lengths corresponding to a more precise place representation. In both analyses, place field sizes/vector lengths showed a unimodal distribution and no indication of discriminable subgroups of place cells with small and large place fields (see Author response image 1, D, for Gaussian size estimate and place vector lengths respectively). Moreover, we found that there was no difference in the distribution of place field sizes when comparing baseline and cue-enriched conditions (see Author response image 1, D).

**Author response image 1. sa2fig1:** Relation of place field length/vector length to the participation of place cells in network events. A, place field size determined by a Gaussian fit for the baseline (blue) and cue-rich condition (red). B, Plot of the place field length versus the fraction of network events in which this particular place cell participates for each cell. C, Plot of the place field length versus the fraction of calcium events associated with a network event. D-F, Same as A-C, but with the place vector length measure.

We are happy to include this in the publication if the reviewer thinks it helpful, but we feel that since we cannot subdivide the place cells into clearly distinct subpopulations, this does not add much to our main conclusions.

What is the proportion of synchronous events that incorporate at least one place cell and/or one speed cell?

We have calculated this and it is on average 23±9 and 33±11% of the synchronous events that incorporate at least one place cell under baseline and cue-enriched conditions, respectively (n=9 mice). 3±2 and 7±3% of events incorporate at least one speed cell (n=9 mice). Currently we do not report these values in the manuscript, but are happy to do so if the reviewer wishes.

Were place cells with a sharp place field more likely to be reactivated?

This is an interesting question. If place cells with more precise tuning are more likely to be reactivated, then the place field length calculated as above should be inversely related to either the fraction of network events the cell participates in, or to the fraction of calcium events occurring within network events. Similarly, there should be a correlation of the place vector length with these parameters. However, neither the place field size nor the vector length were predictive of network event participation. Place field size or vector length were not correlated with the fraction of network events in which the place cell participated (Author response image 1, E, respectively), or with the fraction of Ca^2+^ events in the place cell occurring within a network event (Author response image 1, F, respectively). Again, we have submitted this as Author response image 1 only, because we did not find place cells to fall into two distinct groups, and because there was no correlation with network event participation, but we are happy to include it in the paper if the reviewer thinks it helpful.

What is the total number of cells imaged in this study? If 300 cells were imaged and the place cells represent 2% of the recorded cells only 6 place cells would be recorded? however at least 40 cells are shown in Figure 3.

We are sorry that we did not supply the total number of GCs imaged. We imaged 1415 GCs in the baseline condition, and 1425 GCs in the cue-enriched condition. In additional experiments with a segmented linear track, we imaged 690 GCs. Specifically, we should note that these numbers refer only to GCs that were active during the recording sessions and thus were detected by the non-negative matrix factorization, not to those GCs that were never active. We have added these numbers to the Results section.

9) In the environment enriched with tactile cues the proportion of place cells but not speed cells increases. Does this increase alone explains the higher proportion of cells per network events or were additional non-spatially modulated cells also included?

The reviewer is right in stating that the place cells are found more frequently in cue-enriched sessions. However, this did not explain the increase in place cell incorporation into network events. When we calculated the fraction of all place cells that participated in network events (with the total number of place cells under each condition set as 100%), we saw a strong and significant increase from 55.42 to 88.46 % of place cells participating (baseline vs. cue-rich conditions, see Figure 3C). Thus, irrespective of the increase in the number of place cells in cue-enriched conditions, the probability of a place cell to appear in a network event is significantly higher. We have tried to make this clearer in the revised manuscript (Results).

This could reflect a global increase in network excitability. In general it would be interesting to know the proportion of active and silent DGCs during movement that are active during synchronous events.

The reviewer is right, and we have looked at this. We have separated the GCs recorded into two groups. The first group contains the GCs that are active during running and rest (run-related), and the second group contains the GCs that were not active during running but only during immobility (rest only in Author response image 2). We found that both in the baseline and cue-enriched condition, the fraction of network event participating cells (blue, NE participation) was not different. Thus, it does not seem as if we can deduce a general difference in excitability between these two populations that affects network event participation. We have not included this in the manuscript, but are happy to do so if the reviewer wishes.

**Author response image 2. sa2fig2:** Network event participation of two groups of GCs separated according to their locomotion-related behavior. A, Run-related: GCs that are active during both running and rest, Rest only: GCs that were not active during running but only during immobility. Fraction of network event participating cells in blue, fraction of cells not participating in orange.

10) How do authors explain the high lap to lap variability of PCA trajectories in the dentate gyrus when compared to CA1? Is it due to the low proportion of place cells in the DG or to the use of uncued treadmill?

Firstly, as stated above, we would like to note that our proportion of place cells is not extraordinarily low when normalized to those cells active during locomotion, as commonly done in other publications. Secondly, as described above, we have now added data from mice running on a more commonly used segmented linear track, which also does not have more place cells than our cue-enriched condition.

In all three types of linear tracks, we found a similar, high lap to lap variability also in the segmented linear track, apparent in the PCA trajectories (Figure 7—figure supplement 2A-C). In contrast, CA1 neurons measured under the very same conditions showed a much lower variability (Figure 7—figure supplement 2D). This behavior thus not only occurred in the non-cued belt, but was a very consistent phenomenon across three different types of linear track in DG. In all cases, the behavior was clearly distinct from CA1. We have quantified this phenomenon across all laps in a session by plotting the first 10 components of the PCA across laps for all conditions. In this depiction for CA1, as well as the three different versions of the linear track used for DG experiments, it is clear that strong periodicity for each round is observed in CA1, but much less so in all DG experiments (Figure 7—figure supplement 2E-H). To quantify this phenomenon across animals, we have performed an autocorrelation for all experiments in the four conditions (examples shown in Figure 7—figure supplement 2I-L, averages in Figure 7—figure supplement 2M-P). This also clearly illustrates the phenomenon. If we quantified the autocorrelogram peaks at δ of one lap (corresponding to the first order peak of the autocorrelogram), the difference between CA1 and DG was also clearly apparent (Figure 7—figure supplement 2Q).

These results show that the differences between CA1 and DG are robust, and are unlikely to be a consequence of the features of the linear track. We believe that this difference is a potentially interesting phenomenon in itself. One interpretation of this finding would be that the dentate gyrus is able to represent slightly different, successive rounds in a different way, this itself being a potential manifestation of the pattern separation capabilities of this structure. We cannot prove that this is the case (one of the reasons why we did not focus on this in the prior version of the manuscript), but are now briefly and conservatively discussing these data. We note that a related finding has been obtained for more extended timescales of days recently in a Biorxiv paper published by the group of Thomas Oertner, where they show that even after extensive training in the very same environment on successive days, different sets of dentate granule cells were activated every day (Lamothe-Molina et al., 2020). Albeit at a different time scale, this suggests that the dentate gyrus may discriminate between the same environment presented at different times.

We indeed think that a high variability of place coding between laps is strongly correlated with the probability of detecting place coding cells using a comparison against shuffled data. We would think that the more variable coding is at the population and single cell level across laps, the more unlikely it will be for cells to pass criterion as a place cell.

This high variability complicates comparisons between network synchronous activity and population activity during movement. Indeed, if a network event is similar to population activity during a given lap it will be different from population activity during subsequent laps. We would like to have more precisions on the selection of movement periods and synchronous activities to perform these comparisons. Did the authors compare the activity between a period of movement and the synchronous activity on a lap by lap basis?

The second, related request of the reviewer was to compare the locomotion related activity during individual laps with the synchronous activity of individual NEs. Indeed, the PCA results presented above would suggest that individual network events can be similar to population activity during individual laps, as the reviewer states. We performed such analyses, first calculating PCAs for locomotion related activity for each lap on the linear track on the one hand, and for the network events on the other hand. We then tested which of these individual, lapwise comparisons are significant compared to shuffled datasets (as done for the bulk comparisons initially presented). Examples of these comparisons are shown in the heatmaps in Author response image 3, B. These analyses have revealed that the activity during individual laps is similar to activity during network events, and these similarities have an episodic character, as expected.

We then quantified for each network event to how many lapwise activity patterns the similarity is higher than chance and vice versa (i.e. for each lapwise activity to how many network events it is similar, Author response image 3, D, respectively). Even though, as seen in Author response image 3, D, in individual examples there might be some differences between baseline and cue-enriched conditions, this proved not to be the case in averaged data (Author response image 3, F).

We are of course willing to include this in the manuscript, but felt that this does not add too much to the main conclusions of the paper. We think that – as stated above – this finding is interesting in itself, implying that individual network events may replay population patterns during individual laps. However, without a behavioral protocol that enables us to say that mice have successfully perceived the identity of an individual lap, or a certain number of laps (as i.e. in a protocol that requires mice to count the number of laps it has run), we cannot really prove this. We have therefore included these data as Author response image 3 only, but again, are very willing to include these panels in the paper if the reviewer thinks it makes sense.

**Author response image 3. sa2fig3:** Lapwise similarity of population activity during locomotion with individual network events. A, B, The matrix shows similarity values for the PCA-based comparison between population activity during all network events (x-axis), and during all individual laps. Nonsignificant comparisons are dark blue. A: Baseline example, B: cue-enriched example. C, D, Quantification of the amount of similarity. Panel C shows for individual network events to how many lapwise activity patterns the similarity is higher than chance. Panel D shows for individual lapwise activities to how many network events it is similar. Plots in C, D correspond to data shown in A, B. E, F, as in C, D, but x-axis normalized and averaged (n = 9 mice, 1 session per condition).

11) Population activity during locomotion is more similar to network events than expected by chance but this is unsurprising given that the network architecture morpho-functional properties of dentate granule cells etc… will impose constrains on observed neural dynamics. A useful comparison would be with CA1 where synchronous network events that replay place cells sequences have been thoroughly described. What is the measure of similarity between synchronous events and population activity in the CA1 region? How does the similarity index in DG compares to that in CA1?

We have done the same comparison as we have done for DG for CA1. We have found that the three PCA-based measures we show for the DG comparisons, also show significant similarities between network events and activity during locomotion in CA1. These data have been included in the manuscript (Discussion).

Reviewer #2:In this manuscript, Pofahl et al., found synchronized network events in the dentate gyrus during periods of immobility in mice. These synchronized network events occurred in largely separate groups of granule cells and were correlated with increased excitation coming from the MPP. About half of the cells participating in these synchronized events encoded the position or the speed of the mouse during mobility. Increasing the cues along the track increased the number of cells participating in each network event and increase the fraction of place cells in each network event. Using dimensionality reduction methods, the authors found that a significant amount of the variance of running activity is captured by the space represented by the network events. Separately, using optogenetics the authors inhibit the dentate gyrus of mice during the learning of a pattern separation task and show inactivating dentate gyrus during immobility (away from the objects of interest) can impair acquisition of the pattern separation task. Overall, the manuscript is well-written, and the analysis is done to a high standard. I found the synchronized events in DG, and their relation to activity patterns during locomotion, to be particularly interesting.1) There is no strong connection between the belt (linear track) head fixed task and the object recognition freely moving task. First, in the head-fixed task the authors found synchronized events during immobility that appear orthogonal to each other, but the mice were not engaged in any pattern separation task and so there is no way to deduce the relevance of this phenomenon to pattern separation. Second, using a pattern separation task in freely moving mice, the authors showed that inactivation of granule cells in the dentate gyrus during immobility can impair pattern separation, but they did not show that the synchronized events (found during the head fixed task) occur during this pattern separation task. The most significant connection between the tasks is the occurrence of rest periods in both. It is reasonable to assume that the network events occur during the pattern separation task, but the authors have no proof for this. I strongly suggest that the authors explicitly address this issue in their manuscript and simply state that it is unclear whether the network events occur during the pattern separation test, but assuming that they do occur, then their optogenetic inactivation experiments imply that the network events are involved in the acquisition of the task.Along these same lines, the last phrase of this statement in the Discussion does not appear correct and should be revised: "These events were specifically modified by the environment, showed higher similarity than expected by chance to population activity occurring during locomotion, and were important for formation of memories requiring pattern separation."

The reviewer is correct. It was our aim to test the hypothesis that resting activity might be important for memory formation in a freely moving behavior. We should have stated this clearly. We have revised the presentation of this issue as suggested by the reviewer. We explicitly state that we have not recorded in-vivo in freely moving animals and therefore do not know if network events are present. We also state explicitly that we – because it is impossible to detect sparse network events in freely moving mice to do closed loop stimulation only during network events – are inhibiting network events, but also additional activity of GCs during immobility.

2) In the pattern separation task, it is unclear if ANY inhibition of DG during acquisition can impair learning/memory/preference, or if it is only inhibition during immobility. It seems to me that the following control experiment is missing: inhibit only during locomotion (the same amount of time as during their immobile inhibition experiment).

This is a valid experiment. We agree that it is reasonable to assume that dentate granule cell activity during locomotion, i.e. when the mouse is actively sampling information in the environment, plays an important role in building a memory of this environment. This is why we did not add this control experiment in the first place. We acknowledge that this was an omission. Thus, we followed the reviewer suggestion and repeated the experiments with a new batch of mice expressing eNpHR bilaterally in the dentate gyrus. We tested their performance in the OPS task in trials with inhibition during the entire running period (speed > 5 cm/s). As expected, this also led to a complete loss of preference for the displaced object in the recall trial, indicating that granule cell activity during locomotion is equally important. We have added these data to the manuscript as Figure 8—figure supplement 2.

Reviewer #3:In Pofhal et al., the authors utilize 2-photon calcium imaging of dentate granule cells and MEC input in head-fixed, awake mice to show ensembles of neurons active during immobility are important for the encoding of spatially related information. They find ensembles of neurons that fire during immobility, that these ensembles are largely orthogonal, with a small subpopulation that are shared across ensembles, that network events contain place cells and speed cells, and the ensembles of neurons incorporated into network events is greater in cue rich environments. They go on to show that inhibition of DG GCs during periods of immobility (movement less than 4cm/s) during acquisition of an object place task impairs subsequent retrieval. This paper is potentially interesting and understanding the population-level encoding properties of the DG are certainly of interest to the field. However, the manuscript, as presented, does not provide a strong case that these network events are meaningful. There is no indication what these network events represent, and if they are related to anything significant in terms of dentate physiology or behavior. In addition, I don't see how links can be made between the imaging data and the behavioral data. This and other comments below:1) Figure 1B and 1D should show the same snippet of data in order to be convinced of the fact that these events only occur during immobility. For instance it could be that during movement you simply don't see them.

Figure 1B is designed to give the reader a closer look at representative Ca^2+^ traces at a little higher magnification. This means that we are showing only a short section of data, and a random set of cells (i.e. only very few of the ~500 GCs). Showing network events in the figure format of Figure 1B would mean that we have to cherry pick participating GCs. We are willing to do this, but think that panels such as in Figure 1D that depict all GCs are more useful. To make the reviewer more confident that these events occur during immobility, we now show in Figure 1—figure supplement 3E-J six examples of different sessions and mice, equivalent to those in Figure 1D. Note that in these examples, running speeds are clearly indicated at the bottom to indicate when animals are running and when they are immobile. Amongst these, we also show the session depicted in Figure 1B as an example (corresponding to Figure 1—figure supplement 3J). These examples illustrate that by far most network events occur during immobility.

We are afraid we do not understand the comment stating that we simply might not see network events during movement. We had rigid quantitative criteria for detection of network events, applied to both resting and running episodes, which result in the clear finding that network events mainly occur during immobility. We have extended the manuscript, now including an additional two types of shuffling analysis (see Figure 1—figure supplement 3B-J, and comments to point 2 of this reviewer).

Figure 1D should show all events, not only the synchronous ones, (maybe highlight which cells the algorithm has delineated into a synchronous ensemble) to justify the need to focus on the synchronous ones that happen during immobility.

We have done as requested, and have included the non-network-event activity as gray datapoints in Figure 1D, as well as in Figure 1—figure supplement 3E-J. The network events are color-coded, as in the previous version of the manuscript.

2) The chance level for synchronous events is very generous, as it does not take into account differences in overall activity during network events. The authors should try shuffling to check for synchronous events using time bins of equal activity levels; rather than choosing from a random time bin, time bins should be chosen from those of similar activity level of the population. In other words one needs to control for mean population activity. This would increase confidence that the patterns of synchronous events in the real data still more frequent than chance. Alternatively, a null hypothesis could be generated by simulating a group of neurons with an underlying, varying overall network activity, and make each cell fire independently at random times given that underlying network activity. You will have some neurons that fire together just by chance. So is this chance still lower than what they observe? While the network events may still be real, these would be more rigorous tests.

The purpose of the study was to identify synchronous activity patterns (defined as cells being coactive within a 200 ms window). This corresponds with our imaging study to 1±1 frames. During such very short time windows, finding synchronously active GCs by chance would very rare in the absence of coordinated network , due to the extreme overall sparseness of firing in the dentate gyrus. In fact, network events themselves, if they occur, constitute a substantial fraction of the cells active at that particular time bin, and one could even define a network event as a transient, coordinated increase in firing rate across multiple neurons. Thus, taking either surrogate activity with activity levels identical to those during network events themselves, or shuffling simply using time bins with equal activity to network events would necessarily lead to false non-detection of network events.

We agree with the reviewer, however, that the shuffling analysis should take into account the general level of activity in the population during behavioral episodes, this point is well taken. We had originally included both episodes of locomotion and immobility in the shuffling analysis, thus controlling for levels of population activity across both behavioral states. However, the reviewer is right in that there could be a difference in average activity levels between locomotion and immobility (as suggested by Figure 1—figure supplement 2C, even if the difference was not significant). We therefore repeated the shuffling analysis but shuffled event times for locomotion and immobility periods separately (Figure 1—figure supplement 3C). This shuffling approach also robustly detected network events (Figure 1—figure supplement 3C, examples in Figure 1—figure supplement 3E-J, shuffled data in right panels depicted in red, real data in green).

We furthermore shuffled the data by randomly shuffling the event times for every individual cell (as presented in the previous version of the manuscript). This conserves the firing frequencies of individual cells, but disrupts any temporal correlations. We understand this to satisfy the reviewers request that we make ‘each cell fire independently at random times given that underlying network activity’. These data are depicted in Figure 1E, as well as in Figure 1—figure supplement 3B, with the six individual examples in Figure 1—figure supplement 3E-J (shuffled data in right panels depicted in gray, real data in green, note that shuffled data curves are superimposed and not easily visible).

Finally, we present a third shuffling approach where traces were randomly shifted with respect to each other. This maintains within-cell correlations of firing (i.e. episodes of higher frequency firing), but reduces between-cell correlations. This is shown in Figure 1—figure supplement 3C (averages, green line real data, red line shuffled data). In the representative examples in Figure 1—figure supplement 3E-J, this shuffling approach is shown as red lines in the rightmost panels.

Thus, three different shuffling approaches clearly show detection of network effects that are significantly more common than expected by chance. We would consider this to provide a high level of evidence that network events do not arise by chance.

3) Clustering analysis: This should be assessed with silhouette scores, not mean-r only. In addition " we focused only on those granule cell Ca^2+^ events that occurred within network events", why only during network events? In addition, how was the number of clusters chosen, was there any cross-validation performed?

Determining the optimal number of clusters in a data set is a fundamental issue, for which at present there are no definitive solutions. In general, the methods though can be subdivided into (i) direct methods, which optimize some criterion, such as the within cluster sum of squares or silhouette score, or (ii) statistical testing methods which compare the data against a null hypothesis, such methods for instance include gap statistics.

We have – after much thought – used a method related to the second class of statistical testing methods. As stated in the supplementary data, we first generated Pearson’s r for all cell combinations in a particular imaging session. We then identified clusters of correlated cells using agglomerative hierarchical cluster trees. Clusters were combined using a standardized Euclidean distance metric and a weighted average linkage method. Clusters were combined until the mean of the cluster internal r-value reached a significance threshold. Importantly, the significance threshold was defined by creating a null-distribution of r-values from shuffled data sets. We had shown this in Supplementary Figure 4C and D, of the original manuscript (now Figure 4B, C) where the r-threshold is indicated for that particular example. Thus, the number of clusters was defined based on a specific r value generated from shuffled data from the same session (we hope this answers the last question in this section). We consider this type of analysis a very objective approach, because it allowed us to obtain a precise number of significant clusters at a particular, defined chance level. Moreover, it is a way to base the number of clusters on a statistic relative to a null distribution.

The reviewer has asked for alternative ways of analyzing clustering. We had indeed considered different types of analysis for these data, and have performed K-means and hierarchical clustering, assessing this both with silhouette scores and with within cluster sum of squares measures. When using such methods, indeed we see that in many sessions, increased clustering results in systematic changes in silhouette scores (or cluster sum of squares, see Author response image 4). Currently commonly used methods such as the elbow method considers the total cluster sum of squares as a function of the number of clusters, and then selects an ‘optimal’ number of clusters so that adding another cluster does not lead to a large improvement in the cluster sum of squares. One criticism of this approach is that such estimates of cluster numbers are frequently ambiguous (Tibshirani et al., 2001). This can also be observed in our data, shown in Author response image 4, in which the general magnitude of the optimal cluster number can be broadly inferred, but it is hard to derive a specific number (see Author response image 4 for silhouette score, Author response image 4 for sum of squared distances). We also have used a gap method, which defines the number of clusters relative to a null distribution based on shuffled data using the same clustering approach. Here the optimal number of clusters would correspond to a maximization of the gap value between the shuffled and real dataset. Also, for this approach, however, it is often difficult to derive a clear, unambiguous estimate for a maximum, and hence, the cluster number. We have therefore opted for the described variant of the gap statistic method, which compares against a shuffled dataset, but – as stated above – allows us to derive a precise cluster number at a defined chance level. We hope that the examples we show in the Author response image 4 perhaps make the reviewer more confident that our analysis results in cluster numbers that are in good agreement with those that would be generated from the inflection (or elbow) of the plots of silhouette score vs. cluster numbers (see vertical red lines for our estimate in Author response image 4).

We would maintain that this analysis is rigorous, in particular in view of the fact that we are using it to compare cluster structures in two conditions (baseline and cue-enriched), in which these analyses were carried out in an identical fashion. If the reviewer is not satisfied by these considerations, we would be willing to remove the quantitative descriptive data that is secondary to the rigorous cluster identification (i.e. the description of cluster sizes, etc.). We do not think that these are essential to the main conclusions of our manuscript, although we do think that this correlative structure is interesting and deserves description.

Regarding the question why we focused on network events, we were interested if in this specific type of synchronous activity, a substructure emerges, as seen previously (Malvache et al., 2016). We had explained that we had observed a repeated activation of GCs in network events (see also Video 1), and that we were curious if it is specific correlated sub-ensembles of GCs that are repeatedly recruited in network events.

**Author response image 4. sa2fig4:** Cluster size determination with different methods. A, Silhouette method. For our data, measured silhouettes do not form global or local maxima at certain cluster numbers, but form plateaus around certain values. Usually our determined value (orange line) fell within this plateau. B, Elbow method. Usually our method delivered values that were close to what came closest to an elbow. C, Gap method. Steps in gap value at certain cluster sizes were usually at our or close to our determined value.

4) If cells are tuned so much to speed (Figure 2E) and have almost 0 activity during immobility (as in the heat map), what activity are activity is being analyzed in the network event? In other words, it's a bit confusing to consider how speed cells are involved in immobility.

We see the reviewers point. The correlations indicate that speed cells have a significant correlation of DF/F with speed. The reviewer is right that activity is very low during immobility. However, it is still the case that individual significant Ca^2+^-events of speed cells are detected in network events. These are much less common though than for place cells. When we analyze the number of network events containing at least one place vs. at least one speed cell, it is on average 23±9 and 33±11% of the network events for place cells under baseline and cue-enriched conditions, respectively (n=9 mice). For speed cells, the participation is much lower, with only 3±2 and 7±3% of events incorporating at least one speed cell (n=9 mice). We still think it is important to report the speed cell participation too. We have included this in the manuscript (Results).

In addition, only ~50% of place and speed cells are recruited in network activity, does it mean that they are recruited randomly, by chance?

We do not entirely understand what the reviewer means by chance. We had already shown that the network events themselves do not arise by chance, by comparing against shuffled datasets (now extended with three shuffling methods in Figure 1—figure supplement 3). This means that the participation of GCs in network events (irrespective of whether they are place cells or not) cannot be explained by random co-occurrence of neuronal activity.

In addition, we would like to clear up that the chance level for recruitment is not 50% (i.e. network vs. non-network participation). Instead, the chance level is determined by the number of network events, and the sparseness of place cell activity.

Irrespective of these considerations, we would argue that the fact that in the cue-enriched condition, we see an increased incorporation of place- but not of speed cells in network events also suggests a specific incorporation of place related information in the network events.

5) In Figure 3, is the enhancement of place and speed cells, and increased network activity due to more neural activity in the enriched environment (Figure 5—figure supplement 1A)? This possibility should be excluded.

Regarding the size of network events, the reviewer is right, this may be related to activity increases. We would suggest that this does not have to be excluded, we are not disputing that changes in neuronal state that modify neuronal excitability influence network events. In fact, the pupil data we show supports such a view. We would note, however, that our detection of network events using shuffled data still detects only network events that are significant beyond the chance level, considering increased general activity levels (see also our extended shuffling analyses in Figure 1—figure supplement 3).

The reviewer is right in stating that the place cells are found more frequently in cue-enriched sessions. This certainly also may be due to increased network activity. However – as we understand it – the reviewer is asking if the increased incorporation of place cells into network events (as shown in Figure 3) is due to enhanced general neuronal activity. We do not think that this is the case. We show that the increase in place cells does not explain the increase in place cell incorporation into network events. When we calculated the fraction of all place cells that participated in network events (with the total number of place cells under each condition set as 100%), we saw a strong and significant increase from 55.42 to 88.46% of place cells participating (baseline vs. cue-rich conditions, see Figure 3C). Thus, irrespective of the increase in the number of place cells in cue-enriched conditions, the probability of a place cell to appear in a network event is significantly higher. The same was not true for speed cells. We have tried to make this clearer in the revised manuscript (Results).

A similar point has also been raised by reviewer 1 (Query 9). We refer to the Author response image 2 for the respective analysis. Reviewer 1 suggested to separate the GCs recorded into two groups. The first group contains the GCs that are active during running and rest (run-related), and the second group contains the GCs that were not active during running but only during immobility (rest only in Author response image 2). We found that both in the baseline and cue-enriched condition, the fraction of network event participating cells (blue, NE participation) was not different. Thus, it does not seem as if we can deduce a general difference in excitability between these two populations that affects network event participation.

6) I found Figure 4 confusing and difficult to interpret. They say in the abstract that they "recapitulate patterns evoked in self-motion". It’s not clear to me where this is shown. Maybe I don't understand this figure, but I was expecting they would do PCA during locomotion and compare to PCA during immobility.

Yes, we did systematically compare two PCAs. To ask if network events recapitulation population activity patters evoked in self-motion, we compared PCAs during locomotion to PCAs during network events. To make this approach clear, we had schematically depicted this approach in Figure 4B. We wrote: ‘After applying PCA to locomotor states in the dentate gyrus, we then also performed a PCA analysis of the population activity in the same region during network events. In order to compare the two sets of PCAs representing population activity during running states and network events, respectively, we first used a vector-based similarity measure. Briefly, we projected the traces recorded during locomotion into the PCA-space representing activity during network events, and tested how much of their variance was captured by them. In this analysis, similarity between both population measures would result in a large fraction of explained variance (Figure 7 in the revised manuscript). The principal components, are, by definition, the directions that capture most of the variance of the data they are computed from, and therefore evaluating the ability of NE-PCs to also capture locomotion-associated variability is an appropriate way to measure the similarity of PCs.

This was the first type of analysis we did, which was explained in detail in the methods. We went on to explain the shuffling procedure used to obtain the null distribution. We also have used the two other published similarity measures that have been used so far to quantify similarity between the PCA bases. First a similarity factor 𝐒PCA as described by Krzanowski (Krzanowski, 1979), and the Eros similarity factor as described in (Yang and Shahabi, C., 2004). These were also described in the methods section. We had perhaps not explained this sufficiently at length in the main text, and have tried to make this clearer by explaining the PCA comparison more extensively in the main text (Results).

I’m also confused by Figure 4E, what is being conveyed here? What is the variance explained in their PCAs? Do the first 3 component explain a good fraction of the variance?

We wanted to use the PCAs to provide an illustration of the neuronal state. It may have been confusing that this figure panel was panel E in the submitted version, it could have thus been perceived as belonging to the comparison procedures described in panels C, D and F. We apologize if this has created confusion. We have renumbered the figure panel (now panel D) to make clearer that this is simply illustrative. The first three components explain 50 % of the variance. We note that for the comparison of locomotion related activity to network events, we chose the number of components such that an equal amount (50 %) of variance in each individual data set was explained.

7) When introducing the behavior, the authors state "Collectively, these data suggest that during immobility, GCs engage in structured ensemble activity that reiterates activity during running at the population level. This suggests that such activity might be important for the formation of hippocampal dependent spatial memories." However, the optogenetic experiment doesn't really test this, as the inhibition is impacting a multitude of things outside of these network effects.

This is true, we thank the reviewer for this comment. We note that the experiment that would be ideal to test the role of network events would be to inhibit only the network events. This would entail two-photon in-vivo imaging in freely moving mice, in combination with closed loop inhibition of granule cells. This is an extremely difficult experiment.

However, the reviewers point stands – we completely agree with the reviewer that our behavioral results should not be overstated. We have rephrased this section in the Results and in the Discussion. We now explicitly discuss the possibility that the behavioral effects could potentially rely also on inhibition of non-network event related activity Discussion.

Also, the authors should plot total immobile time (i.e. total time of DG inhibition) for each mouse, and compare to the discrimination index (do the mice that were more immobile, and thus got more inhibition show worse discrimination?). In addition, a more rigorous control would be inhibition during mobility to show that there is any specificity to the immobility period.

We have done so, and show the plots of total immobile time vs. discrimination index in Author response image 5 for all groups. There was no significant correlation in any group. We are happy to add this to the manuscript if the reviewer thinks it is helpful.

**Author response image 5. sa2fig5:** Discrimination indices are not correlated to illumination time. Plotted are individual discrimination indices for individual sessions of recall trials against the illumination time during the respective session. This is shown for the eYFP control group (left panels) and the halorhodopsin group (right panels) for inhibition during acquisition trials (Upper panels, corresponding to data shown in Figure 9) as well as inhibition during recall trials (lower panels, corresponding to data shown in Figure 8—figure supplement 3Q).

We have also performed the requested set of experiments with inhibition during mobility. The hypothesis for this experiment would be to assume that dentate granule cell activity during locomotion, i.e. when the mouse is actively sampling information in the environment, plays an important role in building a memory of this environment. We followed the reviewer suggestion and repeated the experiments with a new batch of mice expressing eNpHR bilaterally in the dentate gyrus. We tested their performance in the OPS task in trials with inhibition during the entire running period (speed > 5 cm/s). This also led to a complete loss of preference for the displaced object in the recall trial, indicating that granule cell activity during locomotion is equally important.

Thus, as written in our response to reviewer #2, inhibition during the entire trial inhibits memory formation. We now show that inhibition only during exploration also disrupts memory formation, as was expected. Interestingly though, our initial result shows that inhibition only during immobility also leads to significantly impaired memory performance, equal in severity to inhibition during the entire trial. We have added these data to the manuscript as Figure 8—figure supplement 2.

8) They state that "We did not observe any indication of epileptiform activity in Thy1-GCaMP6 (GP4.12Dkim/J) mice, as already reported," however this should be shown, specifically in the mice with the 3mm-1.5mm cone implant above dentate gyrus. It would be reassuring to the reader to show that these network events are not due to abnormal activity from the preparation.

Throughout our recordings, we have had no indication of epileptiform activity. Epileptiform activity in Ca^2+^ imaging experiments is characterized by a substantial synchronous recruitment of hippocampal neurons, frequently lasting for many seconds. Activity with such properties was never seen in the dentate gyrus of control animals. The network events we see were – in marked contrast to epileptiform activity – extremely sparse, involving on average only ~ 5-6 % of the neuronal population. This clearly sets them apart from any so far described epileptiform activity pattern. The absence of larger scale synchronous activity in all our experiments has now been explicitly stated in the Materials and methods section.

[Editors’ note: what follows is the authors’ response to the second round of review.]

Essential revisions:1) In the first round of review, the primary concern was the link between network events during immobility and the behavior in the discrimination task. In the revised manuscript, the strong claims of the paper are still not strongly supported by the results shown, and the conclusions of the manuscript should be toned down accordingly and caveats clearly described and discussed. In the revised manuscript, the control experiment has now been provided in Figure 8—figure supplement 1, and the reviewers appreciate this. However, Figure 8—figure supplement 1 shows that inactivation of DG during locomotion in the object place discrimination task also impairs learning. This information is important since it suggests that ANY inhibition of DG during acquisition of the task can impair learning and so it is not clear if there is anything unique about the DG activity during resting for acquisition. Perhaps DG is encoding the experience as a single sequence spanning rest and running, and breaking this sequence at any point could then impact learning. This is an important caveat that should be added as a counterpoint to statements such as the following in the Abstract "Using optogenetic inhibition during immobility, we show that granule cell activity during immobility is required to form dentate gyrus-dependent spatial memories."

We have further toned down the statements made. Specifically, we have added the idea that DG is encoding the experience as a single sequence spanning rest and running to the Discussion section. We indeed did not think of this interesting possibility. In the Abstract, we now clearly state that inhibition during locomotion inhibits learning, but that inhibition only during immobility is also sufficient. Due to space issues, we did not elaborate on the reasons for this effect in the Abstract, but have included these points in the Discussion.

Also, in the Discussion, the authors should include a statement such as the following "however, the same inhibition of DG during locomotion in the OPS task also prevents the formation of spatial memories, making it unclear if DG activity during rest plays a role in acquisition that is unique or different from the activity during locomotion". In general, any claims about specificity of network events during immobility to memory operations, or misleading language, should be revised and toned down. The authors should describe the data as they stand: the synchronized events during immobility may be involved in learning; however, any specificity remains unknown and untested here.

We have included a statement to this effect in the Discussion section. We have also carefully revised the manuscript to remove potentially misleading claims.

2) The huge inter-trial lap by lap variability of population activity in DG during running could be due to the lack of salient sensory cues at fixed locations in the belt or reward. In their response to the previous round of review, the authors suggest that activity in DG is structured by laps because a fixed sensory cue (a change in belt width) is present. But this information is absent from the Materials and methods section, and it is difficult to determine if animals take into account this sensory cue. Authors should find some indication either in their data and in the analysis of behavior that this is indeed the case. The high inter trial similarity of PCA trajectories in CA1 and the data showing that part of the belt is overrepresented seems to suggest that this is the case but this point should be discussed and clearly stated in the manuscript to justify the lap by lap analysis.

In the manuscript, we have used three different types of belt: Firstly, an uncued belt, secondly, a belt with spatial cues (cue-rich), and thirdly, a more traditional linear track with three segments comprising completely different sets of cues. We would like to stress that an inter-trial lap by lap variability far larger than that observed in CA1 was seen in a three types of linear track.

In the uncued belt mentioned by the reviewer, the only spatial cue is the belt junction at which both ends of the belt are joined. We indeed did not mention this in the methods and have now added explicitly that there is a belt junction different in texture even in the uncued belt.

As stated above, we see a very large lap by lap variability of population activity in all three types of linear track. We show this in Figure 7—figure supplement 2, where we plotted the weights of the first five principal components against the animal position for all three conditions in the DG, and show the corresponding CA1 data with the segmented belt for comparison. To quantify how similar population activity is across laps, we computed the normalized autocorrelation of the first five PCA-weights from example data with respect to different laps. The autocorrelation showed clear peaks at integer multiples of 1 lap for CA1, which were much less pronounced for all linear track conditions in DG (see Figure 7—figure supplement 2M-P, for autocorrelations from five first components averaged for all animals). We also quantified the average peak value of weight-autocorrelations at a spatial distance of 1 lap. This revealed a significant difference in the omnibus test (ANOVA, F_(1,3)_ = 88.32, p = 2*10^-30^,). Moreover, the CA1 autocorrelations were significantly different from all three DG conditions. We did find more periodicity in the segmented linear track compared to the baseline condition (* Bonferroni posttest p<0.05, *** Bonferroni posttest p<0.001). We have explained this analysis better in the revision. We have transferred the key quantification of this phenomenon in Figure 7—figure supplement 2Q to the main Figure 7D.

We would also like to emphasize that the periodicity shown in Figure 7—figure supplement 2 does not mean that parts of the linear track are over-represented. In fact, place cells relatively regularly tile the linear track. Instead, the autocorrelation merely shows the repetitiveness of population behaviour with a periodicity of 1 lap (i.e. stable place coding from lap to lap).

3) For the orthogonality of network events, authors should indicate the percentage of orthogonal network events that are likely to occur by chance to give readers an idea of the percentage of "true" orthogonal events.

We have added the bar graphs depicting the comparison to shuffled data in Figure 4A as an inset. We also now give the percentage of orthogonal network events in real and shuffled data in the main text.

4) The variability of PCA trajectories in DG should be quantified and compared to CA1. The reviewers would like to have actual numbers in the text for comparison. Also, PCA trajectories in CA1 should be illustrated in Figure 7 for comparison.

The quantification of the autocorrelation analysis presented in Figure 7—figure supplement 2 has now been included in the main text. We have inserted the numbers and statistics in the text as requested. Additionally, we have transferred the key quantification of this phenomenon to the main Figure 7D. We have also included a representative PCA trajectory of CA1 in Figure 7 for comparison as requested.

5) Reviewers would be more convinced about the specificity of the reinstated activity during synchronous network events by a comparison with random network activity during immobility but outside synchronous network events or a shuffling procedure which preserves inter-neuronal correlations such as procedure number 2 in Figure 1—figure supplement 3.

First, we would like to make clear that in shuffling procedure 2 in Figure 1—figure supplement 3, we shifted the entire time series of cells DF/F by a random time value. This is also the shuffling procedure that we performed for the PCA based similarity measures. Thus, all non-random activity timing between cells is destroyed and cells will no longer be synchronously active at NE timepoints. At the same time, individual cell event statistics will be maintained (i.e. inter-event-intervals). This method thus preserves intra-neuronal correlations and event frequencies, but destroys inter-neuronal correlations. We now better explain this by adding a schematic to Figure 7H-K that illustrates the shuffling procedure.

We agree with the reviewer that in the case of our similarity measures, it is highly relevant how shuffling is carried out. We were not entirely sure that we understood what the reviewers meant with their suggestions for additional shuffling procedures or comparisons. However, it was refreshing and valuable to take the different points the reviewers made as an impetus to rethink what we actually test with different shuffling methods. We consequently added two more shuffling approaches to the paper following a number of additional analyses.

We first tried to follow one reviewer request as we understood it, i.e. to compare locomotor activity with network activity during immobility, but outside network events, with a method that preserves inter-neuronal correlations. One approach we tried was to randomly shuffle NE times (excluding the actual NEs), such that random non-NE times during resting periods are picked. This procedure by definition excludes all activity during NEs, leaving just the activity occurring during immobility outside NEs. Due to this exclusion of a substantial portion of activity, the remaining activity was very sparse, and unfortunately, insufficient to reasonably perform PCAs.

We then stepped back and considered which additional shuffling procedures might be informative. The shuffling approach included in the first submission of this paper, while relevant, does not test if the composition of NEs matters for the similarity between running and NE activity. We therefore implemented two more shuffling approaches that shuffle activity within NEs.

In our second shuffling approach, now included in Figure 7J, we tested if the composition of individual NEs is important. To this end, for each individual NE, we randomly reassigned activity of a given cells activity to a different cell. Thus, NEs have exactly the same number of active cells, but the identity of cells active within them has been randomly changed, and the number of NEs that individual cells participate in will be altered.

In our third shuffling procedure, now included in Figure 7K, we address one reviewers concern that we interpreted as follows: Could it be that morpho-functional properties in the network simply confine activity during run and rest to very specific populations of cells that are always very active? This is a reasonable concern, because the dentate gyrus does contain small subpopulations of cells that are much more active than the rest. If so, then shuffling in a different way would be required. We therefore added a third shuffling method, in which for each cell, we randomly reassigned its NE activity to other NEs. Thus, how many NEs a given cell participates in is maintained. At the same time, NE interactions between specific sets of cells will be altered, although highly active cells that participate in multiple NEs will still be more likely to be co-active in shuffled NEs. If the similarity was driven by such a population of always-active cells, then this shuffling would not disrupt the similarity between running and shuffled NE activity. However, also here NE activity was more similar to running activity than shuffled data.

We think that these data with three different shuffling procedures testing different aspects of similarity between NEs and locomotion related activity show convincingly that there is a similarity at the population level between these two activity types that is larger than expected. Moreover, the two novel shuffling procedures strongly suggest that the cellular composition of network events matters for this similarity.

6) Reviewers recommend deciphering clusters with a different shuffling procedure maybe leading to a higher threshold because some of these clusters do not look convincing in Figure 4 and Figure 4—figure supplement 1.

Indeed, some clusters have a higher intra-cluster correlation coefficient than others. Of note, we have only included clusters into the further analysis when their mean intra-cluster correlation exceeded the 5% significance value which was determined by the shuffling analysis. This means that not all of the clusters appearing along the diagonal axis enter further analysis. We have made this clearer in the text (not only the methods) and have indicated clusters that passed the significance threshold in Figure 4 and Figure 4—figure supplement 1.

**Author response image 6. sa2fig6:** Analysis of clusters with different thresholds. Cluster analysis was done as described in the methods. Significance values of 5%, 3% and 1% are used, determined by shuffling analysis. Clustering examples for eight sessions are depicted on the right. All clusters passing the respective significance threshold are indicated with grey frames. On the left side, we have reproduced the bar and pie charts from Figure 6D and E regarding place and speed cell participation in clusters for all different thresholds.

The reviewers are correct that higher thresholds would lead to smaller clusters with higher intra-cluster correlations. Therefore, we followed the reviewers’ advice and included all analysis using more stringent significance thresholds (3% and 1% instead of 5%) (Author response image 6). As expected, these thresholds result in smaller more convincing clusters in some examples but on the other hand also lead to an increase of inter-cluster correlation in others. However, the general changes observed in place cell participation in clusters are comparable for all different thresholds. We are happy to include this in the manuscript, but since a 5% significance threshold seems to us the most objective approach, and our main findings are robust when changing thresholds, we would suggest leaving it out.

Reviewer #1:Granule cells of the dentate gyrus are numerous and discharge very sparsely despite receiving intense synaptic inputs from the entorhinal cortex. Dentate granule cells are active both during locomotion during which their firing is spatially modulated (place cells) and during immobility periods during which they are synchronously co-activated during large fluctuations in local field potential called dentate spikes in the dentate gyrus and sharp-waves in CA1. In this paper the authors used dual-color bi-photon in vivo calcium imaging in head fixed mice walking on a treadmill to characterize at the cellular level the synchronous activities recorded during immobility in the dentate gyrus and compare them with activities recorded during movement. Mice were running either on an un-cued treadmill (with textures) or a cued-treadmill (with more salient tactile sensory cues). During immobility, they observe co-activation of dentate granule during synchronous network events. These synchronous activities are temporally correlated with increased activity of axonal fibers from the medial entorhinal cortex and correspond to the reactivation of some dentate granule cells active during periods of movement including place cells and speed cells. A number of pairs of synchronous events are orthogonal but synchronous events also reactivate similar clusters of DGCs. The authors then use PCA to compare activity during synchronous events and movement. They report a high lap by lap variability of dentate granule cell population activity (notably compared to CA1 pyramidal cell activity) during movement on the belt and some similarity between synchronous event activity during immobility and population activity during movement. Finally, they observed that inhibiting dentate granule cell activity during immobility impairs performance in an object spatial discrimination task.Synchronous activation of principal cells during awake immobility and sleep has been well described in the CA1 area of the hippocampus and linked to hippocampal dependent spatial memory formation. Much less is known for synchronous activities of dentate granule cells of the dentate gyrus during immobility. Given the sparse activity of these cells two photon calcium imaging seems an appropriate approach to address this question. The study provides interesting characterization of synchronous network activity in the dentate gyrus during immobility. Interpretation of the results is however limited by the high lap by lap heterogeneity of dentate granule cells population activity in the behavioral paradigm used. Furthermore the link between these synchronous network events and memory during an object spatial discrimination task deserves to be strengthened in future studies.1) In the behavioral paradigm used, head-fixed animals are running on belts which can be devoid of any sensory cue or with randomly interspaced cues. Furthermore no reward is provided at a fixed location/distance on the belt. In these conditions it is unclear whether animals get a sense of the dimension of the environment and notably whether they completed single laps around the belt instead of travelling an infinite environment. This point is important because several analyses are performed lap by lap notably the characterization of dentate gyrus place cells in Figure 3 and comparisons between PCA trajectories in Figure 7.

See our comments to point 2 above.

2) Authors use a shuffle distribution to compare population activity during running periods and population activity during immobility-associated synchronous network events and observed that similarity is higher than expected by chance, but this is unsurprising given that the network architecture morpho-functional properties of dentate granule cells etc… will impose constrains on observed neural dynamics. Furthermore, the shuffle method used eliminates inter-neuronal correlations. To decipher the specificity of this similarity, a more convincing comparison would be with activity recorded during immobility but outside network events. Alternatively authors could use a shuffling method that preserves inter-neuronal correlations.

See our comments to point 5 above.

3) Trajectories in PCA space show high inter-trial variability in the dentate gyrus unlike what is observed in CA1. This could result from the lower proportion of spatially modulated cells in the dentate gyrus compared to CA1. This heterogeneity raises a question about the specificity of reinstated activity during synchronous network events.

See our comments to points 2, 4 and 5 above.

4) Authors report preferred reactivations of identical cell clusters during synchronous network events. However, while some clusters illustrated in Figure 4C look convincing (like the red cells' cluster) others appear not very convincing (like the orange cells' one in Figure 4 C). Authors could use a more stringent shuffling procedure to strengthen cluster detection.

See our comments to point 6 above.